# ACCURATE DIFFERENTIAL OPERATORS FOR HYBRID NEURAL FIELDS

## ABSTRACT

Neural fields have become widely used in various fields, from shape representation to neural rendering, and for solving partial differential equations (PDEs). With the advent of hybrid neural field representations like Instant NGP that leverage small MLPs and explicit representations, these models train quickly and can fit large scenes. Yet in many applications like rendering and simulation, hybrid neural fields can cause noticeable and unreasonable artifacts. This is because they do not yield accurate spatial derivatives needed for these downstream applications. In this work, we propose two ways to circumvent these challenges. Our first approach is a post hoc operator that uses local polynomial-fitting to obtain more accurate derivatives from pre-trained hybrid neural fields. Additionally, we also propose a self-supervised fine-tuning approach that refines the neural field to yield accurate derivatives directly while preserving the initial signal. We show the application of our method on rendering, collision simulation, and solving PDEs. We observe that using our approach yields more accurate derivatives, reducing artifacts and leading to more accurate simulations in downstream applications.

## 1 INTRODUCTION

Neural fields are neural networks that take spatial coordinates as input and approximate spatial functions such as images (Sitzmann et al., 2020), signed distance fields (Park et al., 2019), and radiance fields (Mildenhall et al., 2020). With the recent development of *hybrid* neural fields, which modulate the neural network using features from a feature grid, these neural fields can now be trained quickly (Müller et al., 2022; Sara Fridovich-Keil and Alex Yu et al., 2022; Chen et al., 2022) and can approximate large-scale 3D structures such as entire cities (Xiangli et al., 2021; Tancik et al., 2022; Peng et al., 2020; Takikawa et al., 2021).

At first glance, these neural fields accurately represent large, complex spatial signals. However, a closer look reveals several artifacts in the *derivatives* of the represented signal (computed with automatic differentiation, as is standard practice). For example, Figure 1 shows the erroneous derivatives obtained from a grid-based neural field trained to represent the signed distance field (SDF) of a 3D shape. These errors cause significant artifacts when the neural fields are used in established rendering (Takikawa et al., 2021) or simulation pipelines (Chen et al., 2023a) which heavily rely on accurate derivatives. If neural fields are to succeed as a general representation for spatial signals in a variety of applications, we need to address and mitigate such artifacts.

What causes these artifacts in the derivative? The key problem is that neural fields are trained to approximate the *signal* itself – there is no guarantee on the quality of approximation of the signal derivatives. Furthermore, since the neural fields are usually designed and trained to reproduce high-frequency details, they have high-frequency components and are thus liable to have high-frequency noise (albeit of low magnitude). Taking derivatives directly will amplify these high-frequency signals resulting in significant noise (Section 3.1). To obtain a better differential operator, we need to remove this high-frequency noise before computing the derivative.

In this paper, we propose a new approach to compute accurate derivatives on pre-trained hybrid neural fields. Our approach takes inspiration from classical signal processing where derivative computation is typically done on a smoothed version of the signal to avoid amplifying high-frequency noise. Our key idea is to replace direct derivatives of the neural field with derivatives of a *local* low-

Mesh normals AD gradients Ours

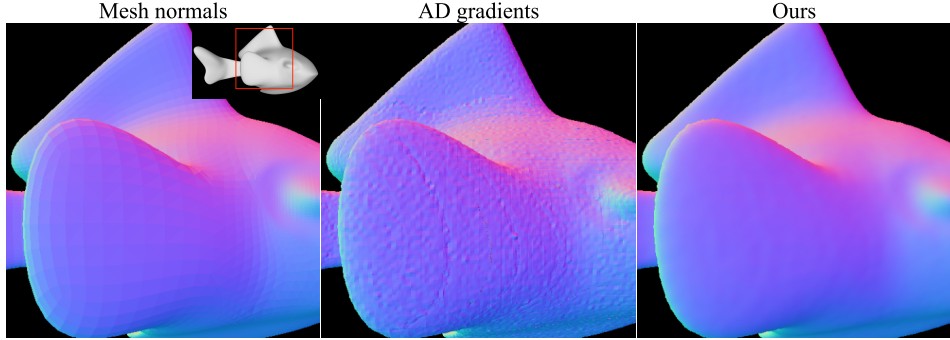

Figure 1: **Noisy gradients in hybrid neural fields**. Normal maps of *Blub the fish* (Stein) (inset), using gradients queried from its hybrid neural SDF using automatic differentiation (AD) and our approach. Naively using AD gradients as surface normals leads to noisy artifacts. Our method is able to alleviate these artifacts.

degree polynomial approximation: $\frac{\partial}{\partial x} F_\Theta(x) \approx \frac{\partial}{\partial x} \sum_{i=0}^{d} a_i x^i$, where parameters $a_i$ are obtained by least-square optimization of a set of labels randomly generated from a neighborhood $N(x)$. These low-order polynomials are easy to optimize efficiently and effectively remove high-frequency noise.

While this approach yields accurate derivatives for off-the-shelf neural fields, it requires that downstream pipelines be changed to use our new derivative operator. To avoid altering downstream pipelines, we need an alternative approach that updates the neural field itself to remove the error in autodiff derivatives. We address this need with a second approach that *fine-tunes* the pre-trained neural field in a self-supervised manner, using the accurate derivatives from the first approach as a training signal. Concretely, we minimize the difference between the autodiff gradients of the neural field and the derivatives obtained from the local polynomial approximation while ensuring that the initial field is preserved.

Our experimental results show that our new derivative operator is yields more accurate derivatives than automatic differentiation (autodiff), reducing errors in gradients by **4**× and errors in curvature by **20**×. It also outperforms other alternative derivative operators, such as finite difference stencils. We also show that our fine-tuning approach yields similar improvements in derivative accuracy without affecting the fidelity of the original neural field. Lastly, we demonstrate that these improvements substantially reduce artifacts in downstream rendering and simulation applications. Thus our proposed methods open the door for using hybrid neural fields in a large set of downstream applications.

**Contributions.** Our overall contributions can be summarized as follows: (1) We identify the issue of inaccurate autodiff derivatives in a given pre-trained hybrid neural field and point out its relationship to high-frequency noise. (2) We propose a local polynomial-fitting operator to improve the accuracy of neural field derivatives (3) We also propose a fine-tuning approach to improve the quality of autodiff derivatives of hybrid neural fields.

## 2 RELATED WORK

**Neural Fields.** Neural fields are neural networks approximating spatial fields given coordinates as input (Yang et al., 2021; Xie et al., 2022). They have been used to represent megapixel images (Martel et al., 2021), 3D shapes in implicit fields (Park et al., 2019; Mescheder et al., 2019; Chen & Zhang, 2019) and radiance fields (Mildenhall et al., 2020; Barron et al., 2021; Verbin et al., 2021). Our work is applicable to hybrid neural fields in all these applications, although our primary evaluation is on SDFs. Typical neural field architectures are multi-layer perceptrons (Sitzmann et al., 2020; Tancik et al., 2020; Mildenhall et al., 2020), but these can be slow to train and may not scale to large scenes with fine-grained details. As such, more current approaches use hybrid representations that modulate an MLP with spatial features stored on a grid (Müller et al., 2022; Takikawa et al., 2021; Yu et al., 2021; Fridovich-Keil et al., 2022; Chen et al., 2022). These hybrid techniques scale well (Peng et al., 2020; Xiangli et al., 2022; Tancik et al., 2022), but we show that they yield noisy derivatives: the key issue we strive to address here. Accurate derivatives are particularly important when neural fields

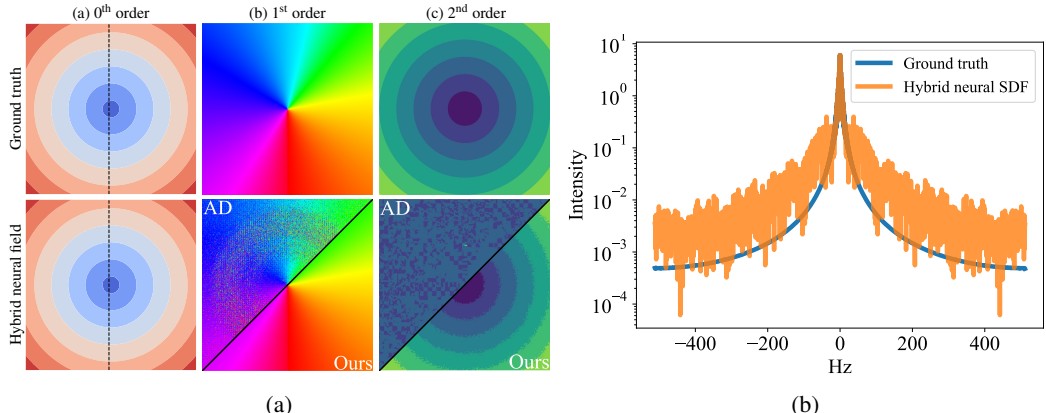

(a)                                                    (b)

Figure 2: **(a) Inaccurate differential operators of neural fields.** Hybrid neural SDF of a circle in 2D. As shown by the comparison with the ground truth, the $0^{th}$ order signal accurately captures the SDF. But the $1^{st}$ and $2^{nd}$-order signals, here shown as the gradient and the radius of curvature (inverse of the Laplacian) suffer from quite a bit of noise. **(b) Fourier spectrum of a hybrid neural SDF.** Computed over a 1D slice (dashed line in (a)) of the SDF of a 2D circle. Note the noisy high-frequency components that are captured by the hybrid neural field.

are used for applications such as rendering (Takikawa et al., 2021; Wang et al., 2021; Zhang et al., 2022) and simulation (Sitzmann et al., 2020; Chen et al., 2023a; Li et al., 2023a; Chen et al., 2023b). Recently, similar to our approach, Li et al. (2023b) used a finite-difference-based regularizer for training hybrid neural fields for surface reconstruction. However, their motivation is to address the training dynamics of hybrid neural fields, instead of removing the high-frequency noise components present in pre-trained neural fields.

**Polynomial-fitting for Shape Analysis.** Polynomial-fitting approaches like Moving Least Squares (MLS) (Nealen, 2004; Cheng et al., 2008; Levin, 2000) have a rich history in 3D shape analysis. Such approaches have found applications in tasks like surface reconstruction from point clouds (Alexa et al., 2003), animating elastoplastic materials (Müller et al., 2004), and learning implicit functions from scattered data (Shen et al., 2004; Ohtake et al., 2003). In this paper, we apply polynomial-fitting to a novel setting of neural fields to solve the important issue of obtaining accurate differential operators. Typically, in past works, given scattered data (point clouds with associated scalar values) as input, approaches like MLS compute fitting planes (or higher-order polynomials) to local subsets of surface points. In essence, the planes/polynomials serve the role of an *interpolant* for the given data (point clouds). In our setting, the neural field *already exists* as an interpolant. But, as we observe in Figure 1, the neural field interpolant does not yield accurate differential operators, and our approach attempts to alleviate this problem. Furthermore, the fitting problem in MLS is different: in MLS, we fit a surface to a set of input points, whereas our approach fits a polynomial to a set of values of the field at any point in its domain.

## 3 METHOD

We assume that we have a pre-trained neural field, $F_\theta$. We further assume that this is a *hybrid field* (Müller et al., 2022), namely, it has a spatial grid of feature vectors in addition to an MLP. The field value at any point is obtained by feeding to the MLP the point location as well as a feature vector obtained by interpolating into the grid. As shown in Figure 1, we observe that the derivatives of such neural fields have noise. Our goal is to come up with an alternative approach that yields accurate derivatives.

To concretize the problem, we focus on neural fields representing 3D shape in the form of signed distance fields (although our final approach is more general). We begin by analyzing why hybrid neural fields yield noisy derivatives and then motivate our approach.

### 3.1 Noisy Derivatives in Hybrid Neural Fields

Why are the derivatives of hybrid neural fields incorrect? We observe that much of the capacity of hybrid neural fields lies in the high-resolution spatial grid of feature vectors. This spatial grid is essential for the neural field to capture fine-grained localized details. Consequently, this spatial grid also determines the high-frequency components of the fitted signal. Unfortunately, this abundance of capacity for high-frequency components means that there are likely many solutions with different high-frequency components that fit the training data well. This in turn can result in noise in the high-frequency components. We observe this noise in practice. Figure 2b compares the spectrum of the ground truth and learned signed distance function (SDF) for a circle in 2D. Note how the learned SDF has higher amplitudes in the high-frequency components.

This high-frequency noise is the source of artifacts in the derivatives. This is because derivative computation accentuates high-frequency noise, scaling it up proportional to the frequency, as illustrated by a sinusoidal signal with frequency $\nu$: $\frac{d\sin(2\pi\nu x)}{dx} = 2\pi\nu cos(2\pi\nu x)$. Thus, even when the high-frequency noise has a very low magnitude, the corresponding noise in the derivative has a much higher magnitude. Figure 2a shows this issue in practice: the same SDF of the 2D circle that we learned earlier provides an extremely noisy gradient when we use automatic differentiation.

**Derivatives and smoothing.** This notion of high-frequency noise magnifying errors in derivative computation is well known in signal processing, and the answer is to use smoothing to remove the high-frequency components. The degree of smoothing can be controlled and corresponds to the scale of the derivative. How this smoothing is done depends on how the signal is represented. For images represented as a 2D grid of pixel values, smoothing can be done by convolving with an averaging filter, and derivatives are typically only computed after smoothing. When 3D shapes are represented as meshes, the mesh automatically represents a smooth version of the signal: each face is effectively a local linear approximation of the surface. Derivatives can then be computed using the face normal.

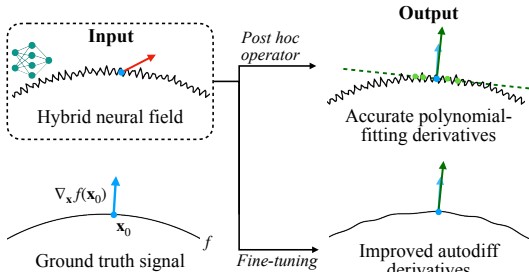

Figure 3: **Problem setup.** Given a pre-trained hybrid neural field with noisy autodiff derivatives, we propose two approaches for accurate derivatives. Our polynomial-fitting operator can be applied in a post hoc manner while our fine-tuning approach directly improves autodiff derivatives of the field.

Unfortunately, no analogous notion of smoothed derivatives exists for arbitrary hybrid neural fields. We address this gap using our proposed approach. Instead of automatic differentiation, our approach first smooths the neural field by computing a local low-order polynomial approximation of the neural field, and then differentiates this polynomial approximation to yield macro-scale derivatives. An overview of this approach is shown in Figure 3.

### 3.2 Local polynomial-fitting operators

Now, we will explain our local polynomial-fitting approach. Given a hybrid neural field, $F_\Theta : \mathbb{R}^m \to \mathbb{R}^n$, and a query point $\mathbf{q} \in \mathbb{R}^m$, we want to compute accurate first-order derivatives of $F_\Theta$ at $\mathbf{q}$. For simplicity of exposition, we choose $n = 1$ here.

First, we sample points $\mathbf{x}_i, i = 1, \ldots, k$ from a local neighborhood $N(\mathbf{q})$ of the point $\mathbf{q}$. We query the neural field to obtain corresponding outputs $y_i = F_\Theta(\mathbf{x}_i) \ \forall \mathbf{x}_i \in N(\mathbf{q})$. We then use these values to fit a local linear approximation $y \approx \hat{F}_\Theta(\mathbf{x}; \mathbf{q}) = \mathbf{g}^T\mathbf{x} + b$ using simple least squares:

$$\hat{\mathbf{g}}, \hat{b} = \arg\min_{\mathbf{g},b} \sum_{i=1}^{k} (\mathbf{g}^T\mathbf{x}_i + b - y_i)^2 \tag{1}$$

Our estimate of the derivative is then $\hat{\nabla}_\mathbf{x} F_\Theta(\mathbf{q}) = \nabla_\mathbf{x}\hat{F}_\Theta(\mathbf{x}, \mathbf{q}) = \hat{\mathbf{g}}$.

We can extend the same approach to the case of vector fields ($n > 1$), where $\mathbf{g}$ is replaced by an $m \times n$ estimate of the Jacobian, $\mathbf{J}$.

**Local neighborhood selection.** Different sampling schemes can be considered in order to select a local neighborhood around the query point $\mathbf{q}$. However, in our experiments, we found that sampling from a Gaussian distribution centered at $\mathbf{q}$, $\mathcal{N}(\mathbf{q}, \sigma)$ worked best for us. The standard deviation, $\sigma$ controls the amount of smoothing that we do at a particular point. The number of neighbors sampled, $k$ is another hyperparameter of our method and controls the variance of the operator that we compute. We discuss how we select these hyperparameters in detail in our experiments (Section 4).

**Hessian & Laplacian.** To compute second-order differential operators like the Hessian or Laplacian, instead of fitting a linear polynomial through the local neighborhood of $\mathbf{q}$, we fit a quadratic surface. Specifically, for scalar fields, we minimize:

$$\sum_{i=1}^{k}(\mathbf{x}_i^T \mathbf{H} \mathbf{x}_i + \mathbf{p}^T \mathbf{x}_i + q - y_i)^2 \tag{2}$$

Ideally, since the Hessian is symmetric and $\mathbf{H}$ is our estimate for the Hessian, we want $\mathbf{H}$ to be symmetric. We therefore reparametrize $\mathbf{H}$ as $\mathbf{H} = \mathbf{D} + \mathbf{U} + \mathbf{U}^T$ where $\mathbf{D}$ is a diagonal matrix and $\mathbf{U}$ is an upper-triangular matrix. Once we obtain $\mathbf{H}$, we can also obtain the Laplacian ($\Delta F_\Theta$) as the trace of $\mathbf{H}$.

Given any pre-trained neural field with similar high-frequency noise, our operators can be applied to it in a post hoc manner to obtain accurate differential operators from the field. However, they do not alter the weights of the neural field, essentially acting as "test-time" operators.

**Comparison to alternatives.** Our approach computes a derivative by sampling points locally and fitting a local polynomial approximation. However, one might consider other alternatives:

1. Instead of automatic differentiation, which yields the instantaneous derivative, we can compute derivatives using finite differences. However, this amounts to sub-sampling the signal without smoothing, which will cause aliasing and thus, inaccuracy in derivatives, as demonstrated in the experiments (see Section 4).

2. A mesh also computes a local polynomial approximation, so we could convert the neural field to a mesh using Marching Cubes. However, extracting a mesh with Marching Cubes can be quite expensive, especially for applications like physical simulation where each simulation step may require gradient queries from an evolving signal (see Appendix D).

### 3.3 FINE-TUNING PRE-TRAINED HYBRID NEURAL FIELDS

The post hoc operator we describe above can be used to effectively query accurate differential operators from a given neural field. However, to use it, every downstream application must be altered to allow for our new operator. Unfortunately, for many applications, autodiff remains the prevalent way to obtain gradients from neural networks. Hence, we propose a method to update the neural field directly so that autodiff yields accurate gradients for hybrid neural fields.

Concretely, given a pre-trained neural field, we propose to fine-tune it to improve the accuracy of the differential operators obtained using autodiff. Let us denote the pre-trained neural field and the resulting neural field by $M$, $F_\Theta$ respectively. $F_\Theta$ is initialized with the weights of $M$. We fine-tune $F_\Theta$ using the following loss function:

$$\mathcal{L}_{ft}(\mathbf{x}_0; \Theta) = \underbrace{|F_\Theta(\mathbf{x}_0) - M(\mathbf{x}_0)|^2}_{\mathcal{L}_{\text{con}}} + \underbrace{||\nabla_\mathbf{x} F_\Theta(\mathbf{x}_0) - \hat{\nabla}_\mathbf{x} M(\mathbf{x_0})||_2^2}_{\mathcal{L}_{\text{grad}}} \tag{3}$$

Here, $\mathcal{L}_{\text{con}}$ denotes the consistency loss which ensures that the output of $F_\Theta$ matches the pre-trained Neural Field, $M$. $\mathcal{L}_{\text{grad}}$ denotes the gradient loss that tries to align the autodiff gradient of $F_\Theta$ with accurate gradient estimates obtained by applying the operator $\hat{\nabla}_\mathbf{x}$ on $M$. In our experiments, we use our polynomial-fitting gradient operator to obtain $\hat{\nabla}_\mathbf{x} M$.

Note that this fine-tuning process is orthogonal to any kind of smoothed gradient operator. Our polynomial-fitting gradient for $\hat{\nabla}_\mathbf{x}$ is just one of the ways we can perform this fine-tuning. We can

similarly use other approaches to compute accurate gradient estimates. In fact, in our experiments, we find that even less accurate estimates, like those obtained from finite differences, can suffice to effectively regularize the fine-tuning.

## 4 EXPERIMENTS AND RESULTS

We first evaluate the accuracy of our proposed operator and then evaluate the result of our fine-tuning approach. For both sets of experiments, we use shapes from the FamousShape dataset (Erler et al., 2020). We pre-train an Instant NGP model (Müller et al., 2022) to learn the SDF of each shape. We evaluate the estimates of surface normals (first-order operator) and mean curvatures (second-order operator) by comparing them to surface normals and mean curvatures obtained from the provided meshes of the shapes (which we regard as ground truth). For surface normals, we compute the mean L2 error, mean angular error in degrees (Ang), and the percentage of points having angle error below $1°$ (AA@1) and $2°$ (AA@2). For mean curvature, we use the rectified relative error (RRE) used by past works for evaluating curvature estimation (Guerrero et al., 2018; Ben-Shabat & Gould, 2020). We report metrics averaged over all evaluated shapes (detailed results for each shape are in Appendix C). For the detailed experimental setup, please refer to Appendix B.

**Choosing $\sigma$ and $k$.** As discussed in Section 3.2, our polynomial-fitting operators also requires $\sigma$ and $k$ values as hyperparameters. In our experiments, we always choose $k = 256$, as choosing a larger value minimizes the variance in the estimated gradients. For $\sigma$, its value is dependent on downstream applications. In this case, since we do not have an explicit downstream application, we choose $\sigma$ to have the best consistency with differential operators obtained from the mesh. Specifically,

- For post hoc operators, we perform a telescoping search for the best value of $\sigma$.
- For fine-tuning, we train an ensemble of models with different values of $\sigma$ and select the value that yields the best autodiff gradients after fine-tuning.

To demonstrate the effect of our hyperparameters, we perform a quantitative ablation over different values on two shapes: the Armadillo and the Stanford Bunny, shown in Figure 4. We also show the qualitative effects of the choice of hyperparameters in Figure 8. The qualitative and quantitative results tell us two things: First, more neighbors are better, and one should choose as large a value of $k$ as computationally feasible; Second, the choice of $\sigma$ depends on the degree of smoothing needed, which in turn depends on the downstream task. However, our accuracy metrics vary smoothly with the choice of $\sigma$, so a coarse telescopic search should suffice.

**Accuracy of operators.** We first evaluate our polynomial-fitting operator by comparing it to automatic differentiation as well as a finite difference baseline.

Table 1 shows our results. Our approach provides more accurate surface normals and mean curvature values from Neural Fields compared to the baselines. In particular, our approach yields $4\times$ reductions in the angular error for the surface normal, and almost $20\times$ reduction in error for Mean curvature. Our approach also yields higher accuracy relative to finite differences.

Table 1: **Operator evaluation**. We compare against autodiff and finite difference baselines on the FamousShape dataset (Erler et al., 2020). We report the performance averaged over the dataset. Note that our approach provides more accurate surface normals and mean curvature than the baselines.

| Method | Surface Normal | | | | Mean Curvature |
|---|---|---|---|---|---|
| | L2 $\downarrow$ | Ang $\downarrow$ | AA@1 $\uparrow$ | AA@2 $\uparrow$ | RRE $\downarrow$ |
| Autodiff | 0.21 | 12.40 | 1.58 | 6.12 | 22.55 |
| Finite difference | 0.07 | 4.20 | 26.86 | 55.22 | 3.67 |
| Ours | **0.05** | **2.80** | **42.92** | **67.90** | **0.89** |

**Improving pre-trained neural fields.** We proposed in Section 3.3 a second approach of fine-tuning the neural field to improve the autodiff derivative estimates. We evaluate two versions of

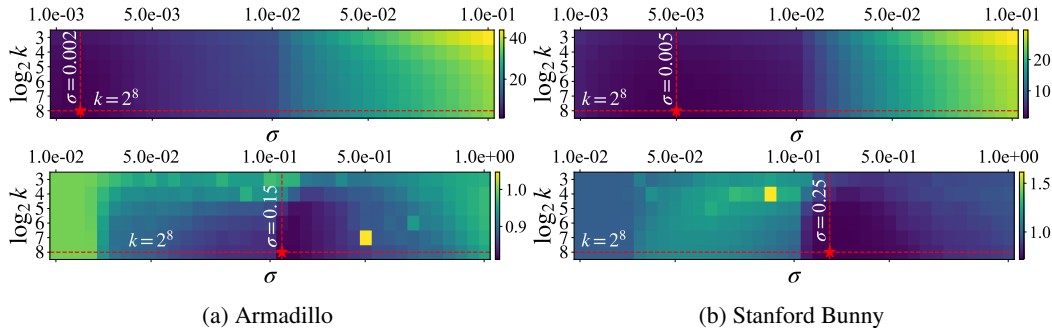

(a) Armadillo                    (b) Stanford Bunny

Figure 4: **Hyperparameter Ablation**. Variation in angle error for normal/gradient (top) and the mean curvature error (bottom) for different settings. $k$ and $\sigma$ refer to the number of neighbors sampled and the size of the neighborhood respectively. $\star$ denotes the best settings.

our fine-tuning approach, one using finite difference-based gradient operators as supervision, and the other using our polynomial fit-based operator. We compare the results from this fine-tuning to the un-finetuned network in Table 2. We observe that fine-tuning improves derivative estimates significantly, with our polynomial fit-based operator providing better supervision. Note that even though we only supervise the gradient, the curvature improves as well. Furthermore, the fine-tuning process preserves the zero-level set of the pre-trained hybrid neural field, as highlighted by minor changes in metrics like Chamfer Distance (CD) and F-Score.

Table 2: **Effect of fine-tuning**. Our fine-tuning approach improves the accuracy of surface normals and mean curvature obtained from automatic differentiation. We compare the performance of autodiff operators before (first row) and after fine-tuning.

| Fine-tuning operator | Surface Normal | | | | Mean Curvature | Mesh Reconstruction | |
|---|---|---|---|---|---|---|---|
| | L2 $\downarrow$ | Ang $\downarrow$ | AA@1 $\uparrow$ | AA@2 $\uparrow$ | RRE $\downarrow$ | CD $\downarrow$ | F-Score $\uparrow$ |
| - | 0.21 | 12.40 | 1.58 | 6.12 | 22.55 | $9.24 \times 10^{-4}$ | 93.07 |
| Finite difference | 0.08 | 5.14 | 21.16 | 46.63 | 3.15 | $9.35 \times 10^{-4}$ | 90.24 |
| Polynomial-fitting | **0.05** | **3.19** | **33.60** | **60.24** | **2.39** | $9.28 \times 10^{-4}$ | 92.28 |

## 5 APPLICATIONS

We now demonstrate the impact of our improved derivatives on downstream applications.

### 5.1 RENDERING

In rendering, accurate surface normals (which correspond to the gradient of the SDF) are needed to estimate how light will reflect off a surface (Shirley & Marschner, 2009). We show the impact of our improved gradients on the rendering of a hybrid neural SDF representing a perfectly specular sphere, and another representing a perfectly lambertian Armadillo (Krishnamurthy & Levoy, 1996).

For the sphere, we use the analytic SDF and surface normals for the ground truth, while we used a mesh as reference for the Armadillo. The sphere was lit with an environment map, and the armadillo with a light source from behind the camera. We use sphere tracing to compute the first ray intersection from the camera with the zero-level iso-surface. Subsequently, we queried the network to obtain gradients using automatic differentiation, finite differences, our post hoc polynomial-fitting operator, and autodiff gradients obtained from a network that was fine-tuned with our operator.

Figure 5 presents our results. As predicted, for the supposedly smooth sphere, as well as the Armadillo, we observed severe surface artifacts using gradients from automatic differentiation. The finite difference-based post hoc operator is able to tackle noise to an extent but still leads to artifacts. On the other hand, normals estimated by our approaches give a much more noise-free image that closely matches the reference.

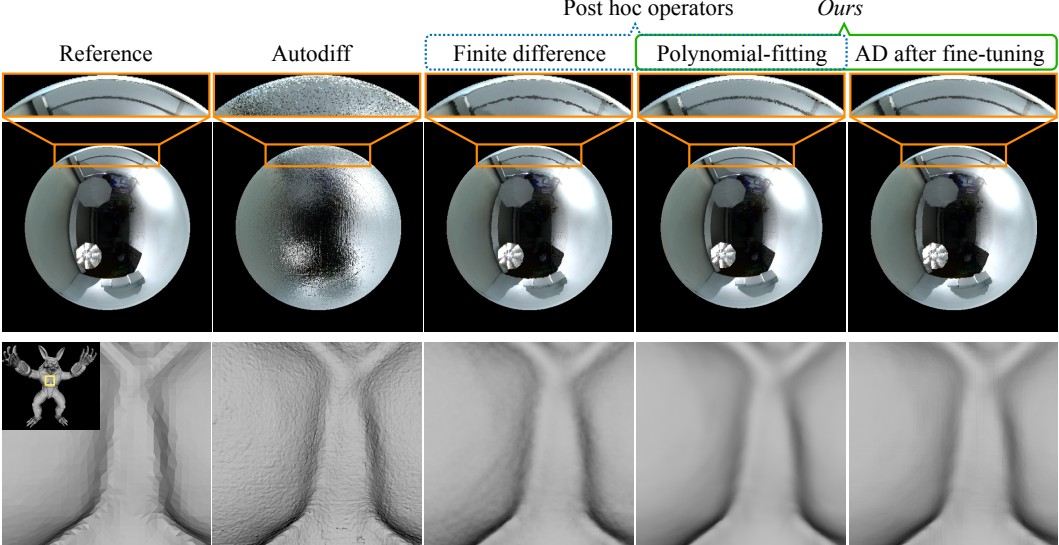

Figure 5: **Accurate Normals for Rendering.** A perfectly specular sphere lighted by an environment map (top) and a diffuse Armadillo (inset) lit by a light source put in front of the object (bottom). In both cases, noisy normals from autodiff lead to artifacts in rendering as shown in the highlighted parts for the sphere and the chest of the Armadillo, that are mitigated by our approaches.

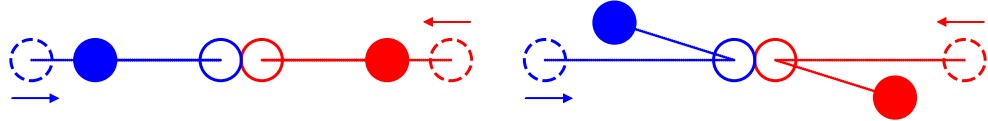

Figure 6: **Effect of noisy normals on collision.** We illustrate the effect of noisy normals on collision. Two spheres undergoing perfectly elastic head-on collisions simulated using correct surface normals will re-trace their paths after a collision. However, inaccurate normal estimates from automatic differentiation yield incorrect trajectories after bouncing.

## 5.2 SIMULATING COLLISIONS

When simulating collisions between objects, normals help determine the impulse direction (Catto, 2005; Erleben, 2007). When working with hybrid neural SDFs, we would need to query the normal at the local coordinates of the point of collision to the network. If the normals are inaccurate, this can lead to incorrect object trajectories after the collision.

For our experimental setup, we consider two identical spheres undergoing head-on collision on a plane and simulate their trajectories post-collision. To obtain these trajectories, we use the normal estimates from the two SDFs at the analytical point of contact. We model the collisions as perfectly elastic so that there is no loss of energy. In the ideal case, the spheres should rebound along the line joining the centers with the same velocity, but erroneous normal estimates will lead to incorrect trajectories. Figure 6 illustrates such a simulation, and also shows how things fail when using autodiff gradients to compute normals. Averaged over $10^6$ trials, the mean error obtained from our normals was $0.85°$, compared to $11.51°$ for autodiff normals.

## 5.3 PDE SIMULATION

Recently, Chen et al. (2023a) proposed using Implicit Neural Spatial Representations (INSR) as the spatial representation of the PDE solution instead of explicit spatial discretization. We build upon

their work and highlight that having accurate gradient operators also enables the use of hybrid neural fields for PDE simulation.

We simulate a 2D advection equation, given by, $\frac{\partial u}{\partial t} = -a\nabla_{\mathbf{x}}u$. For the initial condition, we have a Gaussian wave centered at $(-0.6, -0.6)$ with a standard deviation of $0.1$. We choose a constant velocity, $a = [0.25\ 0.25]^T$. For time integration, we use the forward Euler method, given by,

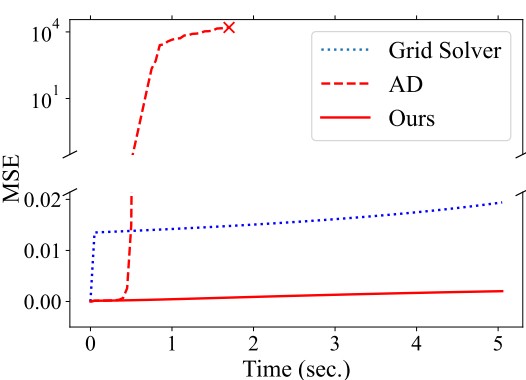

$$u^{t+1} = u^t - a\Delta t\nabla_{\mathbf{x}}u \qquad (4)$$

In our case, the gradient of the initial condition can either be queried directly using automatic differentiation or using our operator. For evaluation, we compare the evolution using our polynomial-fitting gradient operator with the ground truth and the same forward-stepping scheme but with gradients estimated by automatic differentiation (AD). We also show results from a finite difference-based grid solver to show where traditional methods stand. All the methods use a step size of $0.05$, and we run our simulations for $100$ time steps. Figure 7 shows

Figure 7: **Effect of inaccurate gradients in PDE simulation**. Mean squared error (MSE) in a 2D advection simulation, for a finite difference grid solver, autodiff gradients (AD), and our polynomial-fitting approach. The error for AD explodes after the first few seconds and eventually crashes (indicated by ×).

our results. The grid solver accumulates errors over time due to numerical dissipation caused by its spatial discretization. Using our hybrid neural field with autodiff gradients leads to diverging solutions and the evolution collapses after 2 seconds of evolution. Using the same neural field with our operator leads to more accurate solutions at all time steps.

## 6 CONCLUSION, DISCUSSION, AND FUTURE WORKS

This paper studies how to compute accurate differential operators, such as gradient, hessian, and laplacian, for grid-based neural fields. We show that computing these operators via an automatic differentiation scheme will lead to significant artifacts because these derivative operators amplify the high-frequency noise of the model. We tackle this problem via locally fitting a low-order polynomial function and computing a derivative operator on the low-order polynomial function instead. Our method can produce accurate derivative estimates robust to this high-frequency noise. We further propose a self-supervised fine-tuning approach to improve the accuracy of autodiff gradients directly. We show that our method can improve the quality of autodiff derivatives from a pre-trained hybrid neural field. We further demonstrate that our methods improve performance in rendering and physics simulation applications compared to directly using autodiff derivatives for hybrid neural fields.

**Discussion and future works.** The $\sigma$ hyper-parameter required by our approach depends on the downstream task. However, this dependence is not unique to our approach but in fact, is common in other domains where derivatives are computed. For example, in the field of Image Processing, image derivatives have a scale parameter associated with them which depends on the downstream application for which the derivative is being computed. Another analogue, although in a slightly orthogonal setting is the MLS-family of methods for surface fitting on point clouds (Nealen, 2004). These methods often have a "spread" parameter that controls the degree of smoothing of small details. As such, while the sensitivity to this hyperparameter is a limitation, the availability of this knob gives additional control to the downstream application, and can therefore be useful.

Our method also requires randomly sampling a local neighborhood to estimate the parameters, which results in many additional forward passes through the neural fields. This makes our method expensive compared to alternative approaches to compute derivatives. One can potentially speed up our differential operators by sharing the points between different patches or by designing a more efficient sampling pattern. Additionally, if we use our fine-tuning approach then we can amortize this cost and get instantaneous derivatives for subsequent queries.

REPRODUCIBILITY

We present details for the experiments given in Section 4, including data pre-processing, model training, and hyperparameter search in Appendix B. For the results in Secion 4 we present extended results for each shape along with the best hyperparameters in Appendix C. For our quantitative results, we primarily use the FamousShape dataset (Erler et al., 2020) which is publicly available. Appendix E provides details for reproducing the qualitative and quantitative results described in Section 5. We also plan to publicly release the data, code, and models used in our experiments, along with complete documentation upon publication.

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

## A    EFFECT OF HYPERPARAMETERS

As we discuss in Section 3.2, our approach requires two hyperparameters $\sigma$ and $k$, which control the amount of smoothing and the variance of the operator respectively. In Section 4, we also proposed how we can select these hyperparameters, choosing as high a value as possible for $k$ and choosing $\sigma$ depending on downstream applications.

Figure 8 further shows a qualitative example showcasing the effect of $\sigma$ and $k$. We plot the normal maps queried from a hybrid neural field trained on the Armadillo shape. As we can observe, choosing a larger value of $k$ for a fixed $\sigma$ reduces the variance in our operator leading to smoother normals. On the other hand, for a fixed value of $k$, choosing too small a value for $\sigma$ can lead to under-smoothing, while choosing too large can lead to over-smoothing.

## B    EXPERIMENTAL DETAILS

In this section, we provide the implementation details for our experiments described in Section 4.

### B.1    DATASET

We perform pre-training on shapes from the FamousShape dataset (Erler et al., 2020). We filter out shapes with non-watertight meshes or incorrectly oriented normals. This is because non-watertight meshes do not admit a valid SDF and in order to compute the correct ground truth, we require meshes with correct normals. This gave us a set of 15 shapes. We further center the meshes at the origin and normalize them to lie inside the $[-1, 1]^3$ cube.

### B.2    PRE-TRAINING

The inputs for our experiments are the pre-trained hybrid neural SDFs of the shapes. In this section, we present details about how we obtain the pre-trained models.

For our hybrid neural field, we used Instant NGP, keeping the same architecture details as described in Müller et al. (2022). We implemented our models using `tiny-cuda-nn` (Müller, 2021). We follow the same data sampling procedure for training neural SDFs as described by Müller et al. (2022) for training the Instant NGP models. We trained all models for $10^4$ steps using the Adam (Kingma & Ba, 2014) optimizer with an initial learning rate of $1e-3$ and reduced the learning rate by a factor of 0.2 every 5 steps.

### B.3    POST HOC OPERATOR

In this section, we provide details for the hyperparameter selection procedure used for our post hoc polynomial-fitting operator. We used a fixed value of 256 for $k$. For the value of $\sigma$, we selected the best value using telescopic search in two levels: the first sweep is conducted over $10^i : -5 \leq i \leq 1$, after which we zoom in to the interval bounded by the best value, $\sigma_1$ and its best neighbor $\sigma_2$. Assuming without loss of generality that $\sigma_1 < \sigma_2$, we then conduct a sweep over 20 values taken at uniform intervals from $[\sigma_1, \sigma_2]$.

**Baselines.**    We compare our polynomial-fitting operator with automatic differentiation and finite difference for computing surface normals and mean curvatures of the shapes. For automatic differentiation, we directly query the network using PyTorch's (Paszke et al., 2019) automatic differentiation toolkit. For the finite difference operators, we used a centered difference approach, sampling local axis-aligned neighbors of the query point and using them to compute the operator. The finite difference operator had a hyperparameter $h$ for the stencil size. In essence, it gives the size of the finite difference grid cell, if we were to set up a global grid for computing finite differences . We selected this hyperparameter by sweeping over the set $\{\frac{2}{2^i} : 5 \leq i \leq 9\}$. Here $2^i$ is analogous to the resolution of the global finite difference grid.

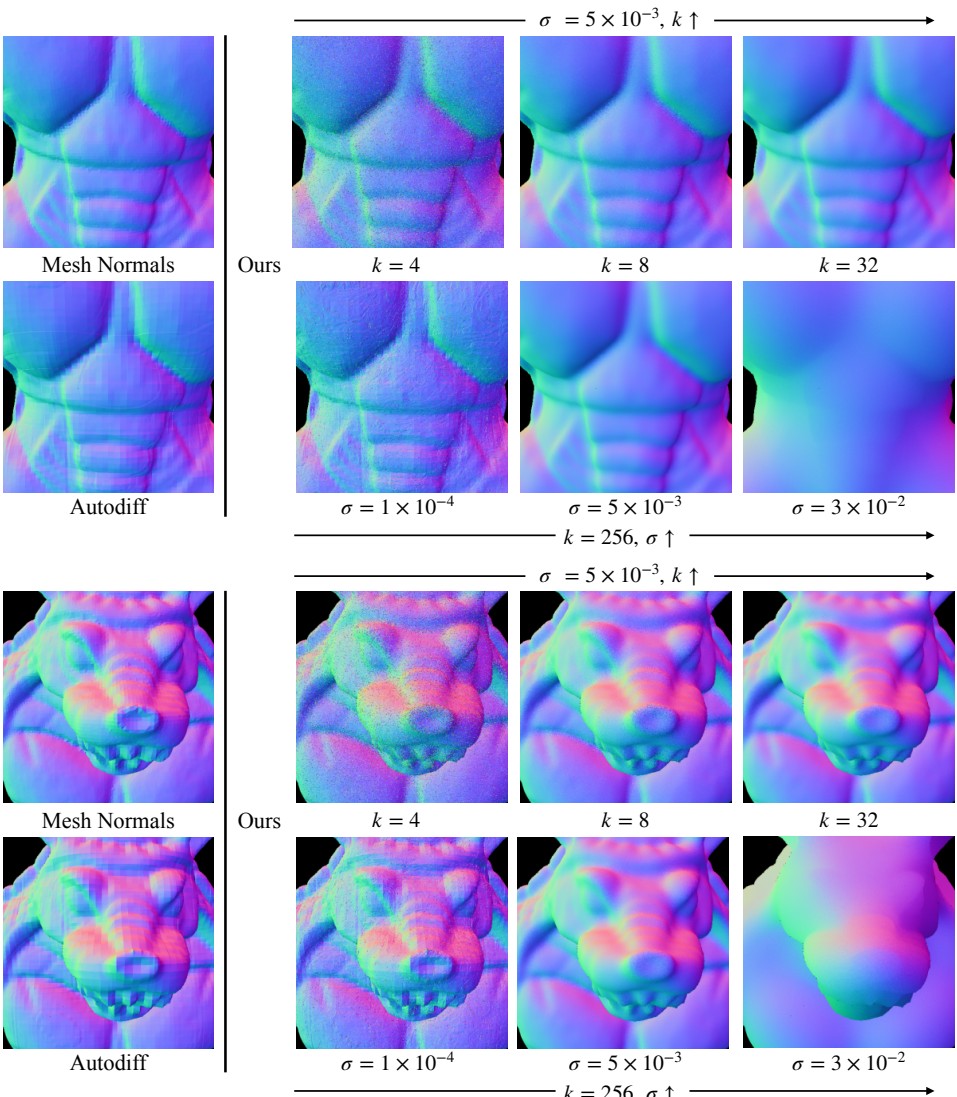

Figure 8: **Effect of hyperparameters.** The performance of our polynomial-fitting operator is influenced by the selected hyperparameter values. We demonstrate this on the Armadillo highlighting this from two different viewpoints, the torso and the head. For a fixed $\sigma$, choosing a larger $k$ can reduce variance in the estimated plane and improve the quality (see the first row for each viewpoint). For a fixed $k$, choosing a large $\sigma$ can lead to over-smoothing, whereas choosing a smaller $\sigma$ can lead to no smoothing at all (second row of each viewpoint).

### B.4 FINE-TUNING

As discussed in Section 3.3, we train an ensemble of models where each model is supervised with a different version of the smoothed gradient operator, $\hat{\nabla}_{\mathbf{x}}$ characterized by the amount of smoothing it imposes. For fine-tuning based on polynomial-fitting derivatives, we ensemble using $\sigma$ values taken uniformly from the interval $[1e-3, 1e-2]$ at steps of $5e-3$. For finite-difference-based fine-tuning we ensemble using stencil sizes from the set $\{2^i : 5 \leq i \leq 9\}$. We fine-tune all models for 4000 steps with a constant learning rate of $2e-3$, using the Adam optimizer (Kingma & Ba, 2014).

## C ADDITIONAL RESULTS

**Accuracy analysis.** In Section 4, we reported the results for the accuracy of our operators. In this section, we provide the full results for the accuracy analysis of our operators and our fine-tuning approach on the FamousShape dataset (Erler et al., 2020). Table 3 shows comparisons between our post hoc operator and the baselines while Table 4 shows how our best fine-tuning approach, i.e., fine-tuning with polynomial-fitting gradients improves autodiff gradients before and after fine-tuning.

Table 3: **Post hoc operator evluation**. We compare our operators on the FamousShape dataset (Erler et al., 2020). $\sigma, h$ indicate the selected hyperparameters for our approach and finite difference (FD) respectively. Note that our approach provides more accurate surface normals and mean curvature than the baselines.

| Shape | Surface Normals | | | | | | | | | | | | | | | | | | | | Mean Curvature | | | | |
|---|---|---|---|---|---|---|---|---|---|---|---|---|---|---|---|---|---|---|---|---|---|---|---|---|---|
| | L2 ↓ | | | Ang ↓ | | | AA@1 ↑ | | | AA@2 ↑ | | | $\sigma$ | $h$ | | RRE ↓ | | | $\sigma$ | $h$ |
| | AD | FD | Ours | AD | FD | Ours | AD | FD | Ours | AD | FD | Ours | | | | AD | FD | Ours | | |
| Angel | 0.12 | 0.04 | **0.03** | 7.14 | 2.30 | **1.50** | 1.93 | 26.47 | **52.09** | 7.50 | 61.66 | **81.56** | 1.5e-3 | 2/512 | | 17.77 | 1.60 | **1.47** | 9.5e-1 | 2/256 |
| Armadillo | 0.13 | 0.03 | **0.02** | 7.27 | 1.83 | **1.18** | 1.79 | 24.69 | **52.52** | 6.88 | 63.93 | **86.84** | 2.0e-3 | 2/512 | | 7.20 | 1.66 | **0.81** | 1.5e-1 | 2/512 |
| Bunny | 0.12 | 0.03 | **0.02** | 6.95 | 1.79 | **1.26** | 2.13 | 42.46 | **67.98** | 8.19 | 78.31 | **88.39** | 5.0e-3 | 2/256 | | 11.38 | 1.37 | **0.72** | 2.5e-1 | 2/256 |
| Column | 0.72 | 0.28 | **0.15** | 46.15 | 16.27 | **8.47** | 0.31 | 0.51 | **4.54** | 1.20 | 2.08 | **15.87** | 3.5e-3 | 2/256 | | 38.15 | 2.54 | **0.88** | 4.5e-1 | 2/128 |
| Cup | 0.12 | 0.02 | **0.01** | 7.06 | 1.24 | **0.88** | 2.02 | 62.26 | **72.20** | 7.93 | 84.37 | **88.66** | 8.0e-3 | 2/128 | | 40.14 | 4.59 | **0.83** | 2.0e-2 | 2/64 |
| Dragon | 0.11 | 0.03 | **0.02** | 6.45 | 1.88 | **1.36** | 2.32 | 29.22 | **54.68** | 8.86 | 69.00 | **86.67** | 2.0e-3 | 2/512 | | 11.13 | 1.46 | **0.89** | 9.0e-1 | 2/256 |
| Flower | 0.26 | 0.08 | **0.06** | 15.21 | 4.50 | **3.39** | 0.63 | 39.12 | **57.18** | 2.52 | 66.99 | **69.30** | 1.0e-2 | 2/128 | | 69.42 | 13.40 | **0.87** | 2.0e-2 | 2/512 |
| Galera | 0.12 | 0.04 | **0.03** | 7.10 | 2.10 | **1.65** | 1.82 | 21.89 | **37.75** | 7.00 | 58.76 | **75.89** | 2.0e-3 | 2/512 | | 6.83 | 1.85 | **0.82** | 2.5e-1 | 2/512 |
| Hand | 0.14 | 0.04 | **0.02** | 8.03 | 2.12 | **1.44** | 1.40 | 19.60 | **39.32** | 5.54 | 55.55 | **79.82** | 1.5e-3 | 2/512 | | 8.74 | 1.27 | **0.86** | 3.5e-1 | 2/256 |
| Netsuke | 0.12 | 0.04 | **0.03** | 7.00 | 2.19 | **1.67** | 1.89 | 21.48 | **41.84** | 7.21 | 56.91 | **74.48** | 2.0e-3 | 2/512 | | 10.24 | 2.40 | **0.82** | 3.5e-1 | 2/512 |
| Tortuga | 0.11 | 0.03 | **0.02** | 6.07 | 1.51 | **1.08** | 2.51 | 46.14 | **63.30** | 9.70 | 80.90 | **89.30** | 3.0e-3 | 2/256 | | 10.43 | 2.36 | **0.74** | 2.0e-1 | 2/512 |
| Utah Teapot | 0.14 | 0.04 | **0.03** | 8.33 | 2.53 | **2.00** | 1.51 | 27.91 | **42.20** | 5.91 | 62.64 | **73.89** | 4.5e-3 | 2/256 | | 38.03 | 7.68 | **0.76** | 3.5e-2 | 2/512 |
| XYZ Dragon | 0.16 | 0.11 | **0.07** | 9.19 | 6.37 | **4.11** | 1.09 | 4.40 | **6.83** | 4.37 | 15.56 | **23.86** | 8.0e-4 | 2/512 | | 5.37 | 2.00 | **0.92** | 1.5e-1 | 2/512 |
| XYZ Statuette | 0.61 | 0.25 | **0.18** | 37.50 | 14.53 | **10.55** | 0.12 | 0.72 | **2.11** | 0.46 | 2.94 | **7.86** | 1.5e-3 | 2/512 | | 43.58 | 9.00 | **0.97** | 2.0e-3 | 2/512 |
| Mean | 0.21 | 0.07 | **0.05** | 12.40 | 4.20 | **2.80** | 1.58 | 26.86 | **42.92** | 6.12 | 55.22 | **67.90** | - | - | | 22.55 | 3.67 | **0.89** | - | - |

Table 4: **Fine-tuning using polynomial-fitting**. Full results for fine-tuning using polynomial-fitting over the FamousShape dataset (Erler et al., 2020). $\sigma$ denotes the hyperparameter value that obtained the best results from the ensemble.

| Shape | Before fine-tuning | | | | | | | After fine-tuning | | | | | | | $\sigma$ |
|---|---|---|---|---|---|---|---|---|---|---|---|---|---|---|---|
| | L2 ↓ | Ang ↓ | AA@1 ↑ | AA@2 ↑ | RRE ↓ | CD ↓ | F-Score ↑ | L2 ↓ | Ang ↓ | AA@1 ↑ | AA@2 ↑ | RRE ↓ | CD ↓ | F-Score ↑ | |
| Angel | 0.12 | 7.13 | 1.92 | 7.54 | 17.96 | 5.32e-4 | 93.47 | **0.04** | **2.06** | **31.74** | **67.25** | **2.01** | 5.38e-4 | 93.45 | 1.5e-3 |
| Armadillo | 0.13 | 7.23 | 1.80 | 6.95 | 7.26 | 1.63e-4 | 96.15 | **0.03** | **1.72** | **31.04** | **69.70** | **1.15** | 1.65e-4 | 96.14 | 2.0e-3 |
| Bunny | 0.12 | 6.98 | 2.08 | 8.11 | 11.36 | 7.26e-4 | 93.26 | **0.02** | **1.37** | **60.74** | **86.52** | **1.07** | 7.09e-4 | 93.35 | 5.5e-3 |
| Column | 0.73 | 46.25 | 0.31 | 1.24 | 39.76 | 2.93e-3 | 85.89 | **0.14** | **8.35** | **4.66** | **16.16** | **2.56** | 2.95e-3 | 85.71 | 3.5e-3 |
| Cup | 0.12 | 7.05 | 2.06 | 7.91 | 40.16 | 3.24e-4 | 94.47 | **0.02** | **1.15** | **60.76** | **84.76** | **4.20** | 3.27e-4 | 89.11 | 1.0e-2 |
| Dragon | 0.11 | 6.46 | 2.31 | 8.81 | 11.03 | 1.99e-3 | 89.97 | **0.03** | **1.90** | **33.25** | **70.87** | **1.34** | 1.98e-3 | 89.96 | 2.0e-3 |
| Flower | 0.26 | 15.20 | 0.67 | 2.52 | 68.78 | 3.40e-4 | 96.48 | **0.06** | **3.32** | **55.34** | **70.17** | **2.09** | 3.47e-4 | 91.49 | 1.0e-2 |
| Galera | 0.12 | 7.10 | 1.82 | 6.93 | 6.81 | 8.37e-4 | 92.29 | **0.03** | **1.97** | **28.11** | **65.63** | **1.20** | 8.35e-4 | 92.31 | 2.0e-3 |
| Hand | 0.14 | 8.02 | 1.39 | 5.53 | 8.78 | 2.64e-3 | 88.08 | **0.04** | **2.10** | **21.30** | **57.41** | **1.56** | 2.67e-3 | 88.10 | 1.5e-3 |
| Netsuke | 0.12 | 7.01 | 1.90 | 7.29 | 10.22 | 1.86e-4 | 96.13 | **0.04** | **2.11** | **26.40** | **61.64** | **1.39** | 1.87e-4 | 96.11 | 2.0e-3 |
| Serapis | 0.11 | 6.57 | 2.21 | 8.55 | 19.73 | 1.18e-4 | 91.79 | **0.03** | **1.65** | **44.96** | **73.21** | **1.61** | 1.18e-4 | 91.73 | 4.0e-3 |
| Tortuga | 0.11 | 6.04 | 2.53 | 9.75 | 10.48 | 3.29e-4 | 96.04 | **0.02** | **1.18** | **61.18** | **87.00** | **1.05** | 3.28e-4 | 96.07 | 3.5e-3 |
| Utah Teapot | 0.14 | 8.29 | 1.50 | 5.92 | 38.41 | 6.23e-4 | 94.30 | **0.04** | **2.06** | **38.10** | **70.07** | **1.45** | 6.31e-4 | 94.13 | 4.0e-3 |
| XYZ Dragon | 0.16 | 9.18 | 1.13 | 4.40 | 5.33 | 9.72e-4 | 90.40 | **0.10** | **5.81** | **4.62** | **16.47** | **1.35** | 9.72e-4 | 90.37 | 1.5e-3 |
| XYZ Statuette | 0.61 | 37.46 | 0.13 | 0.46 | 43.71 | 9.69e-5 | 97.29 | **0.19** | **11.11** | **1.77** | **6.76** | **11.81** | 9.97e-5 | 96.17 | 1.5e-3 |
| Mean | 0.21 | 12.38 | 1.58 | 6.12 | 22.65 | 9.24e-4 | 93.07 | **0.05** | **3.20** | **33.59** | **60.24** | **2.39** | 9.28e-4 | 92.28 | - |

**Analysis on other hybrid neural fields.** We found that the phenomenon of high-frequency noise in hybrid neural fields was not limited to a specific architecture like Instant NGP Müller et al. (2022). We experimented with two other hybrid neural field architectures which showed similar artifacts and inaccurate gradients.

- **Dense grid-based neural field**: A grid-based neural field with dense feature grids, as discussed in Müller et al. (2022). We use a multi-resolution grid with 4 levels, starting from a minimum resolution of 16 up to a maximum resolution of 256.

  Tables 5 & 6 show that our operators can help obtain more accurate gradients than the baselines. This also shows that the artifacts that we observed in the case of Instant NGP were not solely a result of its hash encoding.

- **Tri-planes**: Introduced by Chan et al. (2021). Instead of volumetric grids, they consist of 3 planar grids (one each for XY, YZ, and XZ planes), with a feature embedding residing on each grid point. For a given query point, the features are combined using bi-linear interpolation on each plane and then further summed together. Finally, the feature is passed through an MLP to obtain the output. We used planes with a resolution of 512, feature embeddings of size 32, and an MLP with 2 hidden layers of size 128.

  Results presented in Table 7 show that even on a significantly different hybrid architecture, our operators can provide more accurate surface normals and mean curvatures. At the time of writing, our Tri-plane implementation did not have support for high-order derivatives through autodiff derivatives. Hence, we were unable to compare with autodiff for mean curvature computation or show fine-tuning results.

**Results on images.** We also show the benefits of our approaches on a different modality, specifically images. We train an Instant NGP Müller et al. (2022) model on an image and evaluate its derivatives using our proposed approaches. For pre-training our model, we used a relative L2 loss and trained using the Adam optimizer with a learning rate of 0.01. For fine-tuning, we use MSE loss for $\mathcal{L}_{\mathrm{con}}$, and weighted weighted $\mathcal{L}_{\mathrm{grad}}$ by $10^{-3}$, and trained using a learning rate of 0.02.

Figure 9 shows our results. For reference, we use the derivatives obtained using Sobel filtering, similar to Sitzmann et al. (2020). Firstly, we observe that our fine-tuning approach preserves the initial image, with a minor drop in the PSNR over the pre-trained image. We also compare the accuracy of derivatives using a weighted mean angular error, where the weights are the reference gradient magnitudes. This is because image gradients are usually more important at the regions of high gradient magnitude (the edges). We observe that our post hoc operator gives more accurate gradients than finite differences. We also observe that autodiff gradients obtained after our fine-tuning approach are more accurate than naively applying autodiff to the pre-trained signal.

Table 5: **Post hoc operator evluation on dense grid-based hybrid neural field**. Comparison on the FamousShape dataset (Erler et al., 2020). $\sigma, h$ indicate the selected hyperparameters for our approach and finite difference (FD) respectively. Note that our approach provides more accurate surface normals and mean curvature than the baselines.

| Shape | | | | | | | | | | | | | | | | | | | | |
|---|---|---|---|---|---|---|---|---|---|---|---|---|---|---|---|---|---|---|---|---|
| | Surface Normals | | | | | | | | | | | | | | | | | Mean Curvature | | |
| | L2 ↓ | | | Ang ↓ | | | AA@1 ↑ | | | AA@2 ↑ | | | $\sigma$ | $h$ | | RRE ↓ | | | $\sigma$ | $h$ |
| | AD | FD | Ours | AD | FD | Ours | AD | FD | Ours | AD | FD | Ours | | | | AD | FD | Ours | | |
| Angel | 0.09 | 0.05 | **0.04** | 5.20 | 2.71 | **2.39** | 13.56 | 33.30 | **43.95** | 33.37 | 58.60 | **66.99** | 2.0e-3 | 2/512 | | 4.94 | 3.41 | **0.87** | 4.5e-1 | 2/256 |
| Armadillo | 0.08 | 0.04 | **0.03** | 4.87 | 2.09 | **1.75** | 7.84 | 24.00 | **32.72** | 23.34 | 58.00 | **68.52** | 2.0e-3 | 2/512 | | 2.41 | 1.75 | **1.49** | 9.0e-1 | 2/256 |
| Bunny | 0.07 | 0.03 | **0.02** | 3.78 | 1.73 | **1.26** | 13.10 | 47.91 | **67.93** | 37.00 | 79.02 | **88.29** | 5.0e-3 | 2/256 | | 1.32 | 1.25 | **0.81** | 1.5e-1 | 2/256 |
| Column | 0.27 | 0.21 | **0.14** | 16.07 | 11.98 | **8.33** | 1.96 | 4.09 | **11.48** | 6.96 | 12.64 | **24.65** | 3.5e-3 | 2/512 | | 7.31 | 4.62 | **0.83** | 2.0e-2 | 2/64 |
| Cup | 0.06 | 0.02 | **0.01** | 3.45 | 1.20 | **0.86** | 21.27 | 64.09 | **72.44** | 45.64 | 84.11 | **88.60** | 8.0e-3 | 2/128 | | 1.38 | 1.24 | **0.72** | 2.5e-1 | 2/256 |
| Dragon | 0.08 | 0.04 | **0.03** | 4.56 | 2.25 | **1.76** | 10.73 | 26.83 | **44.20** | 30.92 | 60.64 | **76.86** | 2.5e-3 | 2/512 | | 2.02 | 1.52 | **0.89** | 9.0e-1 | 2/256 |
| Flower | 0.14 | 0.07 | **0.06** | 8.13 | 4.26 | **3.36** | 14.50 | **57.72** | 57.60 | 37.26 | **70.87** | 69.32 | 1.0e-2 | 2/128 | | 4.83 | 3.28 | **0.87** | 2.0e-2 | 2/32 |
| Galera | 0.08 | 0.04 | **0.04** | 4.62 | 2.40 | **2.07** | 10.22 | 23.97 | **32.01** | 29.11 | 55.73 | **65.33** | 2.0e-3 | 2/512 | | 1.37 | 1.70 | **0.82** | 2.5e-1 | 2/512 |
| Hand | 0.09 | 0.04 | **0.04** | 4.93 | 2.55 | **2.03** | 8.24 | 19.34 | **26.81** | 25.71 | 50.25 | **62.64** | 2.0e-3 | 2/512 | | 1.79 | 1.90 | **0.86** | 3.5e-1 | 2/64 |
| Netsuke | 0.08 | 0.04 | **0.03** | 4.56 | 2.26 | **1.99** | 10.12 | 26.22 | **33.08** | 28.77 | 57.91 | **65.06** | 2.0e-3 | 2/512 | | 1.54 | 1.45 | **0.82** | 3.5e-1 | 2/256 |
| Serapis | 0.07 | 0.03 | **0.03** | 4.01 | 1.89 | **1.54** | 17.73 | 39.21 | **48.89** | 36.91 | 67.65 | **75.46** | 4.0e-3 | 2/256 | | 2.53 | 1.98 | **0.93** | 2.5e-2 | 2/128 |
| Tortuga | 0.05 | 0.03 | **0.02** | 2.94 | 1.47 | **1.13** | 17.44 | 50.36 | **62.65** | 45.51 | 80.91 | **87.94** | 3.0e-3 | 2/256 | | 1.32 | 1.60 | **0.74** | 2.0e-1 | 2/512 |
| Utah Teapot | 0.06 | 0.04 | **0.03** | 3.51 | 2.28 | **1.98** | 22.17 | 36.37 | **42.62** | 47.55 | 67.44 | **73.79** | 4.5e-3 | 2/256 | | 2.89 | 0.96 | **0.76** | 3.5e-2 | 2/32 |
| XYZ Dragon | 0.16 | 0.13 | **0.12** | 9.39 | 7.37 | **6.89** | 2.16 | 3.58 | **4.00** | 8.15 | 12.72 | **14.01** | 1.5e-3 | 2/512 | | 1.75 | 2.13 | **0.92** | 1.5e-1 | 2/256 |
| XYZ Statuette | 0.31 | 0.23 | **0.21** | 18.26 | 13.13 | **12.27** | 1.36 | 2.91 | **3.88** | 4.82 | 9.41 | **12.28** | 1.5e-3 | 2/512 | | 7.73 | 10.54 | **0.97** | 2.5e-3 | 2/512 |
| Mean | 0.11 | 0.07 | **0.06** | 6.55 | 3.97 | **3.31** | 11.49 | 30.66 | **38.95** | 29.40 | 55.06 | **62.65** | - | - | | 3.01 | 2.62 | **0.89** | - | - |

Table 6: **Fine-tuning using polynomial-fitting on dense grid-based neural field**. Full results for fine-tuning using polynomial-fitting over the FamousShape dataset (Erler et al., 2020). $\sigma$ denotes the hyperparameter value that obtained the best results from the ensemble.

| Shape | Before fine-tuning | | | | | | | After fine-tuning | | | | | | | $\sigma$ |
| --- | --- | --- | --- | --- | --- | --- | --- | --- | --- | --- | --- | --- | --- | --- | --- |
| | L2↓ | Ang↓ | AA@1↑ | AA@2↑ | RRE↓ | CD↓ | F-Score↑ | L2↓ | Ang↓ | AA@1↑ | AA@2↑ | RRE↓ | CD↓ | F-Score↑ | |
| Angel | 0.09 | 5.20 | 13.64 | 33.45 | 2.51 | 5.33e-4 | 92.87 | **0.06** | **3.62** | **30.12** | **53.67** | **1.34** | 5.35e-4 | 92.50 | 2.0e-3 |
| Armadillo | 0.08 | 4.87 | 7.72 | 23.40 | 1.33 | 1.65e-4 | 95.28 | **0.06** | **3.27** | **14.84** | **39.06** | **1.12** | 1.69e-4 | 95.02 | 2.0e-3 |
| Bunny | 0.07 | 3.78 | 13.11 | 37.07 | 1.39 | 7.25e-4 | 91.00 | **0.03** | **1.59** | **52.63** | **81.11** | **1.02** | 7.15e-4 | 90.67 | 5.0e-3 |
| Column | 0.27 | 16.15 | 1.87 | 6.90 | 4.62 | 2.95e-3 | 84.82 | **0.17** | **9.79** | **4.43** | **13.75** | **2.85** | 2.92e-3 | 73.45 | 3.5e-3 |
| Cup | 0.06 | 3.48 | 21.14 | 45.34 | 7.28 | 3.24e-4 | 84.89 | **0.02** | **1.11** | **63.64** | **84.39** | **2.54** | 3.20e-4 | 78.32 | 8.0e-3 |
| Dragon | 0.08 | 4.54 | 10.74 | 30.85 | 2.01 | 1.99e-3 | 86.31 | **0.05** | **2.72** | **26.48** | **57.89** | **1.21** | 1.99e-3 | 85.95 | 2.5e-3 |
| Flower | 0.14 | 8.14 | 14.34 | 37.25 | 4.72 | 3.40e-4 | 91.50 | **0.06** | **3.40** | **54.03** | **68.48** | **1.50** | 3.46e-4 | 83.98 | 1.0e-2 |
| Galera | 0.08 | 4.62 | 10.08 | 28.87 | 1.36 | 8.41e-4 | 85.93 | **0.05** | **2.93** | **23.18** | **51.34** | **1.11** | 8.36e-4 | 85.87 | 2.0e-3 |
| Hand | 0.09 | 4.95 | 8.16 | 25.72 | 1.72 | 2.64e-3 | 87.68 | **0.06** | **3.28** | **13.77** | **39.75** | **1.28** | 2.65e-3 | 87.59 | 2.0e-3 |
| Netsuke | 0.08 | 4.55 | 10.21 | 29.01 | 1.54 | 1.86e-4 | 92.93 | **0.05** | **2.93** | **20.56** | **47.99** | **1.29** | 1.84e-4 | 92.82 | 2.0e-3 |
| Serapis | 0.07 | 4.01 | 17.48 | 36.62 | 2.50 | 1.18e-3 | 85.15 | **0.03** | **1.97** | **41.59** | **66.92** | **1.37** | 1.17e-3 | 84.85 | 4.0e-3 |
| Tortuga | 0.05 | 2.94 | 17.30 | 45.35 | 1.34 | 3.29e-4 | 93.16 | **0.03** | **1.47** | **51.17** | **80.15** | **1.09** | 3.30e-4 | 93.04 | 3.0e-3 |
| Utah Teapot | 0.06 | 3.52 | 22.07 | 47.55 | 2.84 | 6.22e-4 | 90.70 | **0.04** | **2.19** | **39.15** | **71.07** | **1.08** | 6.24e-4 | 89.93 | 4.5e-3 |
| XYZ Dragon | 0.16 | 9.38 | 2.17 | 8.11 | 1.75 | 9.72e-4 | 89.68 | **0.15** | **8.66** | **2.82** | **10.23** | **1.45** | 9.77e-4 | 89.28 | 1.5e-3 |
| XYZ Statuette | 0.31 | 18.23 | 1.34 | 4.89 | **7.50** | 9.70e-5 | 95.58 | **0.26** | **15.33** | **2.20** | **7.55** | 8.19 | 1.03e-4 | 91.53 | 1.5e-3 |
| Mean | 0.11 | 6.56 | 11.42 | 29.35 | 2.98 | 9.26e-4 | 89.83 | **0.08** | **4.40** | **29.32** | **51.40** | **1.75** | 9.25e-5 | 87.66 | - |

Table 7: **Post hoc operator evluation on Tri-planes**. Comparison on the FamousShape dataset (Erler et al., 2020). $\sigma, h$ indicate the selected hyperparameters for our approach and finite difference (FD) respectively.

| Shape | Surface Normals | | | | | | | | | | | | | | Mean Curvature | | | |
| --- | --- | --- | --- | --- | --- | --- | --- | --- | --- | --- | --- | --- | --- | --- | --- | --- | --- | --- |
| | L2↓ | | | Ang↓ | | | AA@1↑ | | | AA@2↑ | | | $\sigma$ | $h$ | RRE↓ | | $\sigma$ | $h$ |
| | AD | FD | Ours | AD | FD | Ours | AD | FD | Ours | AD | FD | Ours | | | FD | Ours | | |
| Angel | 0.08 | 0.04 | **0.03** | 4.86 | 2.30 | **1.92** | 5.91 | 27.50 | **30.46** | 20.85 | 62.71 | **69.31** | 1.5e-3 | 2/512 | 2.99 | **1.63** | 9.0e-1 | 2/512 |
| Armadillo | 0.09 | 0.03 | **0.03** | 5.35 | 2.00 | **1.48** | 4.04 | 21.41 | **35.38** | 15.00 | 58.19 | **77.42** | 2.0e-3 | 2/512 | 1.23 | **0.81** | 2.0e-1 | 2/256 |
| Bunny | 0.10 | 0.03 | **0.02** | 5.74 | 1.98 | **1.31** | 3.95 | 33.16 | **65.12** | 14.66 | 71.98 | **87.97** | 5.0e-3 | 2/256 | 1.43 | **0.72** | 2.5e-1 | 2/256 |
| Column | 0.38 | 0.24 | **0.14** | 22.87 | 14.18 | **8.31** | 0.70 | 2.06 | **8.21** | 2.68 | 7.60 | **23.31** | 3.0e-3 | 2/512 | 6.07 | **0.95** | 3.0e-2 | 2/512 |
| Cup | 0.09 | 0.02 | **0.02** | 5.20 | 1.31 | **0.90** | 4.86 | 57.47 | **71.78** | 17.38 | 83.76 | **88.42** | 8.0e-3 | 2/128 | 6.32 | **0.82** | 9.0e-4 | 2/32 |
| Dragon | 0.09 | 0.04 | **0.03** | 5.24 | 2.32 | **1.77** | 4.56 | 19.24 | **36.44** | 16.54 | 53.50 | **75.71** | 2.5e-3 | 2/512 | 1.58 | **0.93** | 9.0e-1 | 2/256 |
| Flower | 0.18 | 0.08 | **0.06** | 10.65 | 4.47 | **3.40** | 3.08 | 44.50 | **56.91** | 11.46 | 69.30 | **69.07** | 1.0e-2 | 2/128 | 2.68 | **0.87** | 2.0e-2 | 2/64 |
| Galera | 0.10 | 0.04 | **0.03** | 6.01 | 2.51 | **1.97** | 3.25 | 15.84 | **24.95** | 12.33 | 46.93 | **63.97** | 2.0e-3 | 2/512 | 1.37 | **0.82** | 2.5e-1 | 2/256 |
| Hand | 0.08 | 0.04 | **0.03** | 4.46 | 2.03 | **1.59** | 6.42 | 22.34 | **33.03** | 22.43 | 59.48 | **74.47** | 1.5e-3 | 2/512 | 1.88 | **0.85** | 3.5e-1 | 2/64 |
| Netsuke | 0.10 | 0.04 | **0.04** | 5.92 | 2.52 | **2.05** | 3.47 | 16.26 | **25.08** | 12.94 | 47.23 | **62.22** | 2.0e-3 | 2/512 | 1.55 | **0.82** | 3.5e-1 | 2/256 |
| Serapis | 0.11 | 0.04 | **0.03** | 6.24 | 2.17 | **1.65** | 3.47 | 25.35 | **45.18** | 12.82 | 60.02 | **74.30** | 4.0e-3 | 2/256 | 6.32 | **0.93** | 2.5e-2 | 2/512 |
| Tortuga | 0.09 | 0.03 | **0.02** | 5.32 | 1.76 | **1.23** | 4.36 | 34.08 | **57.78** | 15.83 | 72.24 | **87.04** | 3.5e-3 | 2/256 | 2.86 | **0.74** | 2.0e-1 | 2/512 |
| Utah Teapot | 0.11 | 0.05 | **0.04** | 6.16 | 2.62 | **2.04** | 5.02 | 28.41 | **40.87** | 17.91 | 61.90 | **73.30** | 4.5e-3 | 2/256 | 7.03 | **0.75** | 3.5e-2 | 2/512 |
| XYZ Dragon | 0.22 | 0.13 | **0.12** | 12.80 | 7.52 | **6.96** | 0.72 | 2.40 | **2.43** | 2.87 | 9.12 | **9.22** | 1.5e-3 | 2/512 | 2.78 | **0.92** | 1.5e-1 | 2/512 |
| XYZ Statuette | 0.37 | 0.23 | **0.21** | 22.04 | 13.16 | **11.91** | 0.31 | 1.29 | **1.35** | 1.27 | 5.13 | **5.29** | 1.5e-3 | 2/512 | 15.64 | **0.97** | 2.5e-3 | 2/512 |
| Mean | 0.15 | 0.07 | **0.06** | 8.59 | 4.19 | **3.23** | 3.61 | 23.42 | **35.67** | 13.13 | 51.27 | **62.74** | - | - | 4.12 | **0.90** | - | - |

**Runtime Analysis.** We compare the wall time of our local polynomial-fitting approach with finite difference and autodiff operators. For our operator, we use $k = 256$. We computed the mean and standard deviation of wall-time required by all methods on a single query point, averaged over 7 runs each running 1000 instances of the method. Table 8 summarizes the results. Our operator performs competitively in terms of runtime compared to finite difference and autodiff gradient operators. All these methods were benchmarked using an Instant NGP model (Müller et al., 2022).

Table 8: **Runtime Analysis**

| Method | Runtime (in $\mu$s) |
| --- | --- |
| Autodiff | $1520 \pm 12.9$ |
| Finite difference | $509 \pm 91.1$ |
| Ours | $459 \pm 16.3$ |

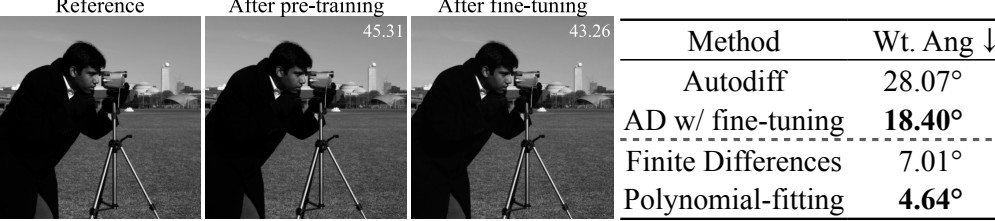

| Method | Wt. Ang $\downarrow$ |
|---|---|
| Autodiff | 28.07° |
| AD w/ fine-tuning | **18.40°** |
| Finite Differences | 7.01° |
| Polynomial-fitting | **4.64°** |

Figure 9: **Results on images.** We show the application of our operators on a hybrid neural field trained to represent an image. For reference, we use the image derivative obtained using Sobel filtering, similar to Sitzmann et al. (2020). We compare the image gradient obtained using our post hoc and fine-tuning approaches with the baselines. For the zeroth-order signal, we show the PSNR (inset) which shows that fine-tuning preserves the initial image. For the image gradient, we show the weighted mean angular error, weighted by the reference gradient magnitude. Applying autodiff after our fine-tuning approach leads to more accurate gradients than direct autodiff. Using our post hoc operator also leads to more accurate gradients than the finite difference-based approach.

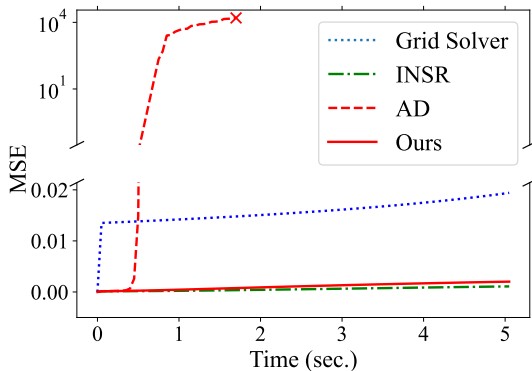

Figure 10: **Comparison to INSR (Chen et al., 2023a)**. While INSR performs better, our approach allows hybrid neural fields to perform competitively, which is not possible when using autodiff gradients directly.

**Comparing PDE simulation with INSR (Chen et al., 2023a).** While we have used the framework of INSR for our PDE simulation experiments, a direct apples-to-apples comparison with INSR is not possible due to INSR utilizing a different architecture (SIREN). As we discuss later (in Appendix F), while SIREN also suffers from inaccurate derivatives, the nature and cause of those accuracies differs significantly from the high-frequency noise that we claim to address. Tackling SIREN's derivative errors would require an altogether different approach that we hope to address in future work.

However, to show how our approach with hybrid neural fields stands relative to a current state-of-the-art approach like INSR, we show a comparison between the errors of our approach compared to INSR in the same setup as discussed in Section 5.3. Figure 10 shows our results. We can see that while INSR performs better, using our approach to compute derivatives with hybrid neural fields allows hybrid neural fields to perform competitively against INSR, which is not possible with autodiff derivatives.

## D  COMPARISON TO MARCHING CUBES

As discussed in Section 3.2, one other alternative for computing derivatives is by directly extracting the mesh using the marching cubes algorithm (Lorensen & Cline, 1987). While mesh extraction with Marching Cubes can take time, this cost can be amortized over multiple queries for derivatives using the extracted mesh. Hence, for a fair runtime comparison to marching cubes, we compare the

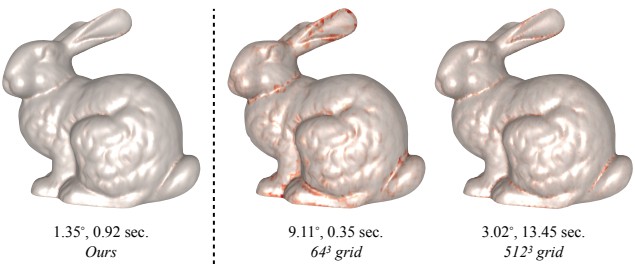

|  |  |  |
|:---:|:---:|:---:|
| 1.35°, 0.92 sec. | 9.11°, 0.35 sec. | 3.02°, 13.45 sec. |
| *Ours* | *64³ grid* | *512³ grid* |

Figure 11: **Marching Cubes for derivatives.** Mean angle error and time required by Marching Cubes to compute surface normals (first-order derivative) on the Bunny shape (Red denotes error). This approach can be expensive (**15×** time) for obtaining accurate surface normals. Reducing the grid resolution can reduce time but trades off accuracy for efficiency (**7×** error). Comparatively, our approach provides accurate normals efficiently.

runtime of our operator with Marching Cubes on a larger point set of size $2^{18}$ sampled uniformly from a 3D shape, in this case, the Stanford Bunny. Since the points sampled may not always lie on the extracted mesh for Marching Cubes, we compute the normals at the closest on-surface point. Figure 11 illustrates how getting comparably accurate derivatives requires running Marching Cubes at a high grid resolution (512) which takes up almost **15×** the time taken by our approach. We can try to save time by running marching cubes at a lower resolution, however, this leads to inaccurate derivatives, resulting in almost **7×** the error incurred by our approach. Thus, getting accurate derivatives from Marching Cubes is quite expensive compared to our approach, and can become increasingly prohibitive in applications like physical simulation, where frequent derivative queries may be required from an evolving underlying signal.

## E  APPLICATION SETUPS

In this section, we describe the details of the experiential setup used in each application described in Section 5.

**Rendering.**  In our rendering experiments (Section 5.1) for both shapes, we used the Instant NGP model (Müller et al., 2022). The pre-training, fine-tuning, and hyperparameter selection were done using the same process as described in Appendices B.2 and B.4 respectively. For our polynomial-fitting operator, we use $\sigma = 0.03$ and $k = 256$ for the sphere and $\sigma = 0.002$ and $k = 256$ for the Armadillo, selected using telescopic search. For the results of the fine-tuning approach, we queried all models in the ensemble and selected the best render after visual comparison. For the finite difference operator, we selected a stencil size of $\frac{2}{32}$ for the sphere and $\frac{2}{512}$ by conducting a sweep as described in Appendix B.3.

**Simulating Collisions.**  For our experiments on simulating collisions (Section 5.2), the hybrid neural SDF of the sphere was a multi-resolution grid-based model like Instant NGP (Müller et al., 2022) but with a dense grid instead of a hash grid. The model had a minimum resolution of 16, a maximum resolution of 128, and consisted of 4 grid levels. For our polynomial-fitting operator, we used $\sigma = 0.03, k = 64$, selected using telescopic search.

**PDE simulation.**  For the PDE simulation experiment (Section 5.3), we used the same model architecture as the collision experiments, with a minimum resolution of 16, a maximum resolution of 128, and 4 grid levels. We modify the code shared by the authors of INSR (Chen et al., 2023a) to solve the 2D advection problem. However, we retain the data sampling and the training strategies used by the authors such as uniform sampling of the domain for training the implicit field, and early stopping during optimization. Our initial condition is a Gaussian pulse, given by:

$$f(x,y) = e^{-\left(\frac{(x-\mu_1)^2 + (y-\mu_2)^2}{2\sigma^2}\right)} \tag{5}$$

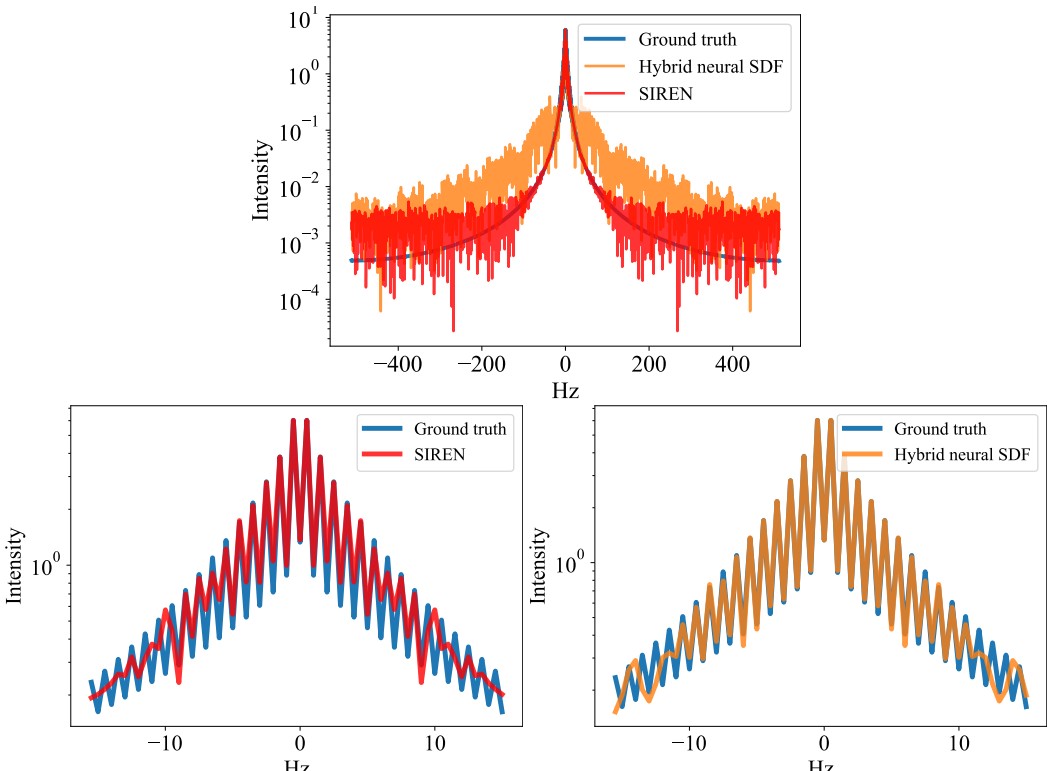

Figure 12: **Fourier spectrum of SIREN Vs. hybrid neural SDF.** Computed over a 1D slice (shown in Figure 13) of the SDF of a 2D circle. Note the lower degree of high-frequency noise compared to the hybrid neural SDF. Further zooming in (bottom row) to visualize the low-frequency components reveals the low-frequency errors in SIREN. Comparatively, hybrid neural SDF more accurately captures the lower frequencies.

where $\mu_1 = -0.6, \mu_2 = -0.6, \sigma = 0.1$. We run our simulations in a square of side length 2 centered at $(1, 1)$. For the boundary conditions, we use the Dirichlet boundary condition, i.e., the field becomes 0 at the boundary, the same as INSR (Chen et al., 2023a) in their 1D advection setting. Other details are shared in Section 5.3.

## F  EFFECTIVENESS ON A NON-HYBRID NEURAL FIELD (SIREN)

While our approaches are not tied to a particular architecture, they can only address the high-frequency noise in neural fields. We do not claim to address low-frequency features. As we illustrated in Section 3, signals learned by hybrid neural fields like Instant NGP (Müller et al., 2022) are abundant in such high-frequency noise.

We also investigated if similar kinds of artifacts arise in non-hybrid networks, specifically SIREN Sitzmann et al. (2020). Our first observation was that even for SIREN, derivatives, particularly higher-order derivatives, suffer from inaccuracies. However, unlike hybrid neural fields, we found that SIREN has a lower degree of high-frequency noise. The errors in SIREN seem to be stemming from low-frequency errors. Figure 12 illustrates this phenomenon.

Using our operators to compute the spatial derivatives of SIREN only helps to a limited degree (Figure 13). The observations on gradient are not very interesting as the autodiff gradient itself for SIREN is quite good and our operator leads to minor improvements. However, when computing the curvature (or the Laplacian), we observe that while autodiff curvatures are quite inaccurate, our operator can recover some reasonable values from the field, but noticeable errors remain. We suspect that while our operator can deal with the high-frequency noise component in the underlying field, it is not able to overcome the low-frequency errors in SIREN.

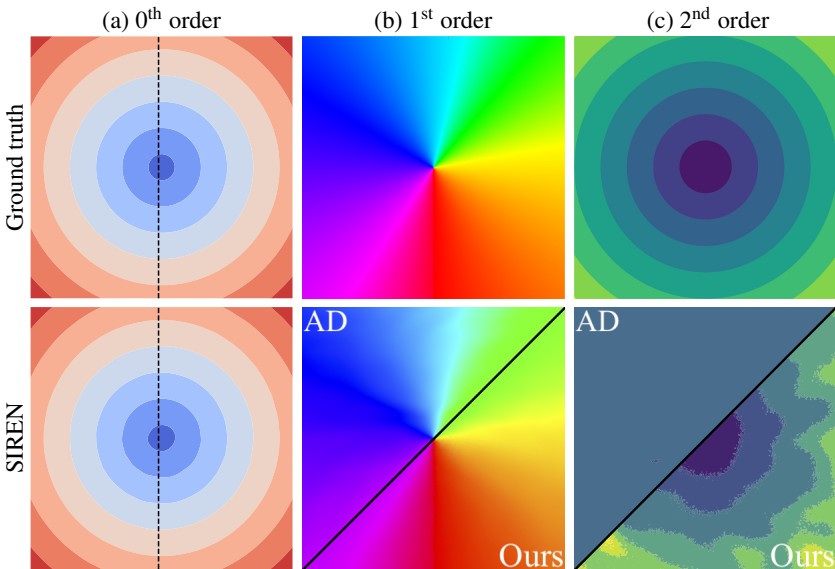

Figure 13: **Differential operators of SIREN.** SIREN trained on the SDF of a circle in 2D. While the first-order operator (spatial gradient) for SIREN is quite accurate, the second-order operator (or the Laplacian) exhibits large errors. Applying our operators shows limited effectiveness, addressing the high-frequency noise in the signal but struggling with the low-frequency errors.

To conclude, our preliminary experiments reveal that neural fields learned by SIREN have a lower degree of high-frequency noise and higher low-frequency errors compared to hybrid neural fields. As a result, while our operators can deal with high-frequency noise, low-frequency errors still result in inaccurate derivatives. Dealing with these low-frequency errors would require an altogether different approach and would be an interesting direction for future work.

