# OpenReview forum: "Accurate Differential Operators for Neural Fields"
_ICLR.cc/2024/Conference — Submitted to ICLR 2024_

### Official Review · Reviewer_6pQ5 · 2023-10-26

**Soundness:** 3 good
**Presentation:** 2 fair
**Contribution:** 1 poor
**Rating:** 3
**Confidence:** 4

**Summary:**

This paper introduces a post-processing method to improve the estimation of first- and second-order derivatives on learned hybrid neural fields (e.g., Müller et al., 2022a), a.k.a. implicit neural representations (INRs). By applying the local polynomial regression, the estimates of these derivatives are smoothed w.r.t the size of the local sampling domain and the number of samples in the local domain. An auxiliary approach is also proposed to fine-tune hybrid neural fields by biassing the estimates via autograd toward the previously proposed smoothed operator.

**Strengths:**

- Quality: The paper offers a range of practical downstream applications to illustrate the utility of the approach for hybrid neural fields. While it may not yield surprising results for most proposed tasks, it does exhibit improvements compared to pre-trained hybrid neural fields.
- Clarity: The approach is lucidly presented, drawing motivation from empirical observations and classical techniques in shape analysis.
- Significance: The approach may be beneficial for downstream applications requiring certain types of neural field architectures.

**Weaknesses:**

- It seems that the paper's claim regarding motivation from empirical observations might be somewhat overstated, particularly when applying it to **all** types of neural fields. The problem of noise overfitting is notably more prevalent in neural fields that heavily emphasize local details but might not be as prominent in other more global architectures like SIREN or Multiplicative Filter Networks. The authors should consider providing additional evidence, particularly in the context of these architectures, to support their claim for all neural fields.
- Novelty: Novelty: The proposed method can be applied to any continuous function, but its contribution from a machine learning perspective is limited apart from adding the fine-tuning loss.
- It seems that the authors modified the neural field architecture used in Chen et al., (2023a) for the application of PDE simulation (sec. 5.3). Will the claim of avoiding prediction error explosion still hold with the original architecture?
- When working with signals that inherently contain high-frequency features, as often encountered in applications like Physics-informed approaches (such as Chen et al., 2023a) for fluid dynamics, it may not be appropriate to employ unified smoothing parameters across the entire domain.
- Minor issues:
  - Please double-check the citations in your paper. There are some instances of:
    - Duplicated citations (e.g., "Chen et al., 2022c" and "Chen et al., 2023a", "Müller et al., 2022a" and "Müller et al., 2022b", "Tancik et al., 2022a" and "Tancik et al., 2022b").
    - Incorrect formatting (Sara Fridovich-Keil and Alex Yu et al., 2022).

**Questions:**

- Does the prediction error explode using the original architecture with SIREN as in Chen et al., (2023a)?
- In Fig. 7, the grid solver gives non-zero loss at the initial time. Is there any issue in the simulator?

---

> ### Author Response · Authors · 2023-11-20
> **Response to Reviewer 6pQ5**
>
> Thank you for your valuable feedback about our work! We have addressed your comments below.
> - - -
> ## Experiments on non-hybrid architectures
>
> We have addressed this in the general response, under “_Experiments on … (SIREN, etc.)_”
>
> We would like to clarify that we only claim to address inaccurate derivatives caused due to high-frequency noise in neural fields. While our approach is not tied to a particular architecture, as discussed, we found such noise to be more prevalent in hybrid neural fields. Non-hybrid architectures like SIREN seem to be plagued mostly by low-frequency errors. We have made edits to the main text to make it clearer and have also added our preliminary analysis on SIREN to Appendix F in the revised paper.
> - - -
> ## Novelty
> As rightly noted by __Reviewer LQfg__, the simplicity of our approach is also its strength, and its effectiveness is supported by our quantitative results. Furthermore, to the best of our knowledge, ours is the first work to highlight the issue of high-frequency noise in hybrid neural fields. We have also not come across any other work that claims to address the issue of inaccurate spatial derivatives using autodiff for hybrid neural fields, like our fine-tuning approach. Nor could we find any previous work that uses polynomial-fitting in the context of neural fields to address inaccurate derivatives.
> - - -
> ## PDE Simulation
> We do not claim that the prediction error would also explode for SIREN, i.e., the architecture used in Chen et. al. We also tried Chen et al.’s approach in our PDE simulation experiment and it did not suffer from exploding prediction error (see Appendix C, under "_Comparing PDE simulation with INSR_" and Figure 10 in the supplementary).
>
> As we discussed in the general response, under “_Experiments on other … (SIREN, etc.)_”, SIREN suffers from a different kind of noise which is mostly low-frequency. While it does still lead to inaccurate derivatives, it does not lead to an error explosion in this case.
>
> We would also like to emphasize that our motivation with the PDE simulation experiment was to merely show that without our approach, hybrid neural fields like Instant NGP lead to exploding errors in applications like PDE simulation, thus limiting their downstream applications. We also observe in Figure 10, that despite minimal additional tuning, using our approach with the hybrid neural field leads to comparable performance w.r.t. INSR, a state-of-the-art approach for this problem.
> - - -
> ## Unified smoothing
> It is indeed true that using a unified smoothing parameter is not ideal when the signal itself may have high-frequency components. However, we would like to point out that our operator is capable of accepting a different smoothing parameter for each query point. We agree that this however does not address the challenge of finding the right $\sigma$ for each point of the field. That would depend on the downstream task as well. But as we discuss in Section 6 (under “_Discussion and future works_”), the dependence of $\sigma$ on the downstream task is not limited to our approach, but in fact, is common in other domains where derivatives are computed, like Image Processing.
> - - -
> ## Formatting Issues
> Thank you for pointing out the formatting issues. We have addressed them in the revised version. Also regarding Fig. 7, we had mistakenly plotted the error from the first time step and omitted the error at t=0. We have fixed this in the revised version.

---

### Official Review · Reviewer_LQfg · 2023-10-31

**Soundness:** 3 good
**Presentation:** 3 good
**Contribution:** 3 good
**Rating:** 8
**Confidence:** 4

**Summary:**

This paper addresses a critical issue in neural fields, namely, the gradient problem, which holds immense significance in computer vision (e.g., 3D reconstruction in NeRF; consider the neuralAngelo paper), geometry processing, and solving PDEs. The authors propose a method to accurately compute differential operators, including gradients, Hessians, and Laplacians, for hybrid grid-based neural fields. This approach effectively mitigates high-frequency noise issues associated with previous hybrid neural field techniques, particularly on high-resolution grids, by locally fitting a low-degree polynomial and computing derivative operators on the fitted polynomial function. The author's approach outperforms both auto-differentiation-based and finite difference-based methods.

**Strengths:**

Clear Problem Formulation: The authors adeptly address the critical gradient problem in neural fields, making a significant contribution to the field and potentially enabling broader adoption of neural fields over traditional representations like meshes and point clouds.

Novel Approach: The proposed polynomial fitting approach is both straightforward and highly effective, tackling issues related to high-frequency noise and artifacts.

Experimental Validation: The authors thoroughly validate their approach across various neural field applications, including rendering, collision simulations, and PDE solving.

Hyperparameter Discussion: The paper provides a very nice discussion of hyperparameters, particularly the parameters 'k' and 'sigma.' The comparison with MLS methods, which also involve hyperparameters (spread), also adds value to the discussion.

Comparison with Previous Methods: The authors conduct a comprehensive comparison with baselines using a variety of datasets, such as the FamousShape dataset

**Weaknesses:**

The primary concern lies in the scalability of this method. Most of the presented results pertain to normal-scale setups. While these results are suitable for the paper's scope, questions arise regarding the method's ability to handle gigascale scenarios, especially with the availability of gigascale NeRF implementations.

**Questions:**

It would be beneficial if the authors could provide more insights into the scalability of their method and its potential applicability to gigascale NeRF scenarios.

---

> ### Author Response · Authors · 2023-11-20
> **Response to Reviewer LQfg**
>
> Thank you for the positive review of our work! We are elated that you agree with our problem formulation, and liked our “novel” approach and our experimental results.
>
> - - -
>
> ## Scalability
> We can break down the scalability question into two parts – post hoc operators and fine-tuning:
> - ___Post hoc operators___: Our post hoc operators act on the pre-trained models. Hence, it is possible to apply our operators directly even on gigascale models.
> - ___Fine-tuning approach___: Our fine-tuning approach can also be applied to gigascale models. Our current formulation of post hoc operators would indeed require an increased training budget owing to the multiple forward passes required for the smoothed gradient computation for supervision. As we discuss in Section 6 (under “_Discussion and future works_”), caching local neighborhoods between nearby points or designing more efficient sampling patterns could be some approaches to address this.
> - - -

---

> > ### Comment · Reviewer_LQfg · 2023-11-23
> >
> > Thank you for your response. This helps with answering my questions regarding scalability.

---

### Official Review · Reviewer_gdRC · 2023-10-31

**Soundness:** 3 good
**Presentation:** 3 good
**Contribution:** 2 fair
**Rating:** 6
**Confidence:** 4

**Summary:**

In this paper the authors tackle issues that arise with hybrid (grid-based) neural fields in the computation of differential operators. They emphasize that automatic differentiations leads to inaccurate and noisy derivatives with a trained hybrid neural field $F$.  Then, they introduce a local-polynomial to approximate the SDF field in the neighborhood of a query point, in order to take the differential operators of the local polynomial instead of that of $F$. This approach can be useful at test time, but requires computation at each iteration. To tackle this issue, the authors propose a fine-tuning that is supervised with smoothed gradients while preserving the represented signal.
They perform some experiments to validate the effectiveness of the local-polynomial and of the fine-tuning, and then show that this increased accuracy in derivative computations has a positive impact on several downstream tasks (rendering, collision simulations, 2d advection).

**Strengths:**

The paper is really well written and enjoyable to read. The angle of the paper is original and the local-polynomial fitting sounds like a simple yet effective idea to smooth out the derivatives of a neural field. The authors experimentally validate their polynomial-based differential operators on the FamousShape dataset, on which they outperform autodiff and finite difference across all metrics. Subsequently, they experimentally confirmed that fine-tuning with polynomial fitting lead to better results than with finite differences.

**Weaknesses:**

All the experiments were conducted with instant-ngp, a particular implementation of a hybrid neural field. Therefore, it is difficult to understand if the problem is systematic across all methods of this class. It would also have been interesting to compare the method with non-grid-based neural fields, such as SIREN, Fourier Features, MFN, etc.

Similarly, apart from the PDE simulation section - which seems to be unrelated to the rest of the paper - all the experiments were done with SDF data. It would strengthen the paper if the method would also apply to other modalities.

**Questions:**

For post hoc operators, how do you find the best value of $\sigma$ ? What is the criterion to choose the $\sigma$ that yields the best autodiff gradients ?

Would the method work with different neural field architectures ? Did you try other grid-based neural fields ? Would it be also useful to apply this method for non-grid-based neural fields ?

Did you try your method on other datasets and with other modalities ?

How do you solve the polynomial fitting ? How long does it take for one query ?

What is the training pipeline for fine-tuning ? How long does it take ?

Could you provide more details on the PDE simulation section ? Can you compare your result with standard INR ?

---

> ### Author Response · Authors · 2023-11-20
> **Response to Reviewer gdRC**
>
> Thank you for your valuable feedback about the paper! We are elated to see that you found the paper “enjoyable to read”, and our idea to be “original” and “effective”.
>
> - - -
> ## Comparison to other hybrid/non-hybrid architectures
>
> We have addressed this in the general response, under “_Experiments on … (SIREN, etc.)_”.
>
> Amongst hybrid neural fields, we found that apart from Instant NGP, even “dense” multi-grid architectures that do not utilize hash embedding, as well as tri-plane neural fields also suffer from similar inaccurate derivatives caused by high-frequency noise that our method can address. For more details, please refer to the general response.
>
> As discussed in the general response, we would like to clarify that we only claim to address inaccurate derivatives caused due to high-frequency noise in neural fields. While our approach is not tied to a particular architecture, as discussed, we found such noise more prevalent in hybrid neural fields. Non-hybrid architectures like SIREN seem to be plagued mostly by low-frequency errors (Appendix F in supplementary) which cannot be corrected by our approach
>
>
> - - -
> ## Experiments on other modalities and datasets
>
> Based on your feedback regarding experiments on other modalities, we have also added some results on neural fields for images. See Appendix C, under “_Results on images_”).
>
> In addition, we also show experiments on PDEs, precisely to demonstrate that our approach is beneficial in other applications apart from improving derivatives of SDFs (Section 5.3). In our PDE experiment, we use our accurate differential operators to compute the self-supervised update shown in Eq. 4 of the paper and compare it to the case when the AD derivative is used.
>
> We would also be happy to test our method on any other specific dataset that you suggest.
> - - -
> ## Hyperparameter details
>
> We have provided details for hyperparameter selection in Section 4 (under “_Choosing $\sigma$ and $k$_”), Appendix A, and Section 6 (under “_Discussion and future works_”).
>
> - - -
> ## Computational Cost
>
> We address this in the general response, under “_Computational Cost_”. For fine-tuning the details of the training pipeline are provided in Appendix B.4.
>
> - - -
>
> ## Clarifications about PDE results
>
> All the details for the PDE simulation experiments are shared in Section 5.3 and Appendix E (under “_PDE Simulation_”). We would be happy to provide any other specific information that is required.
>
> **Comparison to INSR**: (We assume INR in the review is a typo). We have added a plot in the supplementary that compares our approach with the original INSR implementation [A] (Appendix C, under “_Comparing PDE simulations with INSR_”). Our approach allows hybrid neural fields to perform competitively w.r.t. INSR which uses a completely different architecture (SIREN). We would further stress that the primary comparison should be between our approach and AD, because of this difference in representation type from INSR and the reasons noted in “_Experiments on … (SIREN, etc.)_”.
> - - -
> [A] Implicit Neural Spatial Representations for Time-dependent PDEs. Chen et al. ICML 2023

---

> > ### Comment · Reviewer_gdRC · 2023-11-22
> > **Answer to rebuttal.**
> >
> > Thank you for the answers. I will raise my score to 6.
> > However, I still think that the title and the motivation should reflect the capabilities of the method to improve differential operators of  hybrid neural fields. Adding a "hybrid" term before "neural fields" in the title would be more aligned with the claims of the paper.

---

> > > ### Author Response · Authors · 2023-11-22
> > > **Thank you for your feedback!**
> > >
> > > Thank you for your feedback and raising your score! We would be happy to change the title as you suggested to better reflect the problem that our work addresses. We have made this change in the revised version of the paper and will also update it for the camera-ready version.

---

### Official Review · Reviewer_DXqV · 2023-11-01

**Soundness:** 2 fair
**Presentation:** 3 good
**Contribution:** 2 fair
**Rating:** 3
**Confidence:** 4

**Summary:**

In this paper, the authors address the important problem of inaccurate derivatives computed via automatic differentiation (autodiff) from hybrid neural fields. The authors identify high-frequency noise amplification during derivative computation as the core issue. They propose two solutions - a  polynomial fitting operator to smooth the field before differentiation, and a method to fine-tune the field to align autodiff derivatives with more accurate estimates. Experiments demonstrate 4x and 20x reductions in gradient and curvature errors respectively over autodiff.

**Strengths:**

- Identifies a significant limitation of hybrid neural fields that could impede their use in downstream applications requiring accurate derivatives.
- Proposes two concrete solutions that are justified from both theoretical and experimental perspectives.
- Achieves impressive quantitative improvements over baselines.
- Demonstrates reduced artifacts and improved results when using the proposed methods in example applications like rendering, simulation, and PDE solving.

**Weaknesses:**

- **Lack of rigorous literature review**: It seems that the authors have ignored some of the very relevant papers in this domain. The problem of inaccurate and noisy gradients is a well known problem and several people have proposed alternative solutions: (1) DiGS : Divergence guided shape implicit neural representation for unoriented point clouds (2) Given the high cost of dense point cloud capture, can we use sparse point clouds?​ (3) Use a local region prior to generalize to various unseen local reconstruction targets​ (4) NeuFENet: Neural Finite Element Solutions with Theoretical Bounds for Parametric PDEs (5) mechanoChemML: A software library for machine learning in computational materials physics. These are just a few examples of how people try to use other approaches to compute the derivative than, the approach that authors take. While the authors may and probably will try to defend the paper saying that these papers are not relevant, I would argue that they are. For example: [2] uses the nearest neighbors idea to compute SDF and update the derivative (using AD of course). [4,5] use Lagrange polynomials from Finite Elements to evaluate the gradients for backprop, [1] solves the AD noise problem by adding an additional loss for having zero divergence. Each of them are valid alternative approaches to what the authors of this paper are planning to do. A good paper would try to compare with them and ensure that their approach is truly necessary.

- **Computational cost** - The method seems to be completely based on a pre-trained network. Which means, the model is essentially being trained twice.

- **moving target problem** - if the authors use the pre-trained model as a reference, do they assume that pre-trained model is good enough to approximate the properties? Do they consider, updating the weights of the pre-trained model after say 1000 iterations? (just like it is done in DQN-type algorithms?). This could be a great concern if the noisy gradients dont allow the model to converge to a solution at all.

- **Instant-NGP** - It seems that all the experiments from the author are using Instant-NGP as their baseline and working further from there. Naturally, there are issues with Instant-NGP that the multi-resolution hash encoding is not well understood in terms of derivatives. Some recent papers provide some mathematically sound hash encodings for AD (In Neus or Neus++ paper I think). Therefore, doing the experiments using some non-hash-encoding-based methods like SIREN or Nerf (although time-consuming, would be better).

-**Comparisons** - comparison with Finite-Difference (which is already built-in inside Instant NGP) and AD is like a typical strawman approach where you choose several approaches and rigorously test them is missing. I would like to see more than this.


-

**Questions:**

See the weaknesses above.

---

> ### Author Response · Authors · 2023-11-20
> **Response to Reviewer DXqV**
>
> Thank you for your insightful feedback about the paper! We are glad that you liked the significance of the problem and were impressed by the results.
>
> - - -
>
> ## Literature Review
> We appreciate all the pointers, and agree that the problem of inaccurate derivatives appears in many many settings. We will cite these papers in our related work section. However, our work is primarily focused on hybrid neural fields which (a) have a specific kind of high-frequency noise which is exacerbated by derivative computation, and (b) are used commonly enough that these issues will plague many downstream applications. In particular,
> - (1) assumes a SIREN network, and aspects of the method, like initialization strategies, are specifically designed for SIREN.
>
>     - As we discussed in the general response (under “_Experiments on … architectures (SIREN, etc.)_”), the issue of high-frequency noise is not present in SIREN (See Appendix F in the modified paper). Hence comparing with a baseline based on SIREN would not be an apples-to-apples comparison.
>     - Furthermore, the work does not provide evidence that any specific aspects of their method lead to more accurate AD derivatives. Hence, it is not clear why trying to adapt it for hybrid neural fields would help with our motivation.
>     - While they do have a term that minimizes the divergence of autodiff gradients, it is different from our alignment loss.
> - (4) & (5): The methods described in these approaches use significantly different representations from hybrid neural fields.
>      - Our motivation with this work is to improve a well-accepted hybrid neural field like Instant NGP to make it better at providing more accurate derivatives. We could not find previous works that utilize Lagrange polynomials with hybrid neural fields for more accurate derivatives.
>     - Adopting Lagrange polynomials into hybrid neural fields is a really interesting approach, but we feel that this would be a completely different alternative that should be explored in a separate paper. it could be better explored as a future work. This is because it would lead to an altogether different representation with its unique properties.
> - (2) & (3) do not seem to be the right titles of the papers. We would be grateful for a clarification so that we can address them.
>
> - - -
>
> ## Computational Cost
> We have addressed this in the general response (under "_Computational Cost_")
>
> - - -
>
> ## Moving Target Problem
> We generally find that the pre-trained model represents the zeroth-order signal (in our experiments, the shape SDF or the initial value function) accurately. If you mean that noisy gradients would be a problem during pretraining then no, we believe that noisy derivatives with respect to spatial coordinates should not affect training, which relies on derivatives with respect to *model parameters*. That said, one can definitely update the original model after derivative finetuning. We have avoided doing so to keep our approach simple.
>
> - - -
>
> ## Comparison to non-hybrid architectures
>
> We have addressed this concern in the general response,  under “_Experiments on … (SIREN, etc.)_". Addressing the point about NeUS and NeUS++, NeUS does not use hash encodings, and by NeUS++ if the reference is to NeUS2 (kindly correct us if we guessed wrong), then they only propose a more efficient way to compute the second-order derivative through the hash-grid, rather than proposing new more mathematically sound hash encodings. They also do not provide any evidence that their formulation leads to more accurate AD derivatives.
>
> - - -

---

### Author Response · Authors · 2023-11-20
**General Response**

We thank all the reviewers for their insightful feedback! We have addressed common concerns in this general response. Major changes in the paper are colored red for convenience. We would love to hear your thoughts about our rebuttal, including whether it sufficiently addresses your concerns and questions. If you believe that our rebuttal is satisfactory, it would be great if you could consider increasing your score. Any feedback is welcome and greatly appreciated!

## Contributions
We first reiterate our contributions. Our contributions are three-fold:

1. We identify artifacts in the differential operators (e.g., gradients, curvatures) computed from hybrid neural fields such as Instant NGP [A]. We observe that these artifacts arise because even when the neural field approximates the true underlying field well, it has high-frequency noise that is exacerbated by derivative operators.
2. We propose a local sampling and polynomial fitting approach to obtain more accurate differential operators from pre-trained hybrid neural fields in a post hoc manner.
3. To allow downstream pipelines direct access to accurate derivatives through autodiff, we further propose a fine-tuning approach that uses our post hoc operator to supervise autodiff gradients of the neural field, while preserving the original field.

- - -

## Experiments on other neural field architectures (SIREN, etc.)
___Non-hybrid architectures___: As we mentioned above, our approach can address __high-frequency noise__ in neural fields, leading to more accurate derivatives. While our approach itself is not tied to any specific network architecture, we found that high-frequency noise is more prevalent in hybrid architectures. In our preliminary experiments, we found that while non-hybrid architectures, specifically SIREN, are also prone to inaccurate derivatives, the reason for these inaccuracies seems to be different from the high-frequency noise that we observe for Instant NGP. High-frequency noise is of a lower magnitude in SIREN. Instead, low-frequency errors seem to be more abundant. Due to this difference, our method is not well-suited for obtaining accurate derivatives from SIREN. Tackling this issue for SIREN would require altogether different approaches which we hope to address in future work. We have made this clearer in the main text (Section 3) and have also added a note about our preliminary experiments with SIREN in Appendix F.

___Hybrid architectures___: For testing hybrid architectures, we selected Instant NGP to use a standard and well-accepted architecture. However, we have now added experiments with other hybrid architectures, including “dense” grid architectures, i.e., without the hash data structure utilized by Instant NGP, as well as in another hybrid neural field, Tri-planes [B]. We find similar artifacts and demonstrate that our approach can help alleviate them. (see Appendix C, under “_Analysis on other hybrid neural fields_”).

- - -

## Computational Cost

Reviewers DXqV, 6pQ5 had questions about the computational cost of our method.

We have provided a run-time analysis of our post hoc operators in the supplementary (Appendix C), which shows that our post hoc operators perform competitively with the baselines. To answer reviewer 6pQ5’s question, we use Pytorch’s internal least squares solver which internally chooses between LU or QR factorization for computation.

For our fine-tuning approach, in each training step, our post hoc operator computes accurate gradients for the sampled points and then uses them for training the autodiff gradients. The overall training takes 4k steps or ~2 hours. However, we already get ~90% of the reported performance in ~1k steps or ~30 minutes. These times are reported for the standard Instant NGP architecture. Using smaller grids can further improve this time.

Reviewer DXqV further questioned if our approach requires training our model twice. Note that the pre-trained hybrid neural field can perfectly represent the zeroth-order signal (eg, the shape SDF or the initial value function in PDE solving). We only require an additional fine-tuning step to ensure accurate AD derivatives. This is because our approach is self-supervised using the accurate zeroth-order signal learned by the signal. This is a one-time measure, and subsequent AD queries do not require training again. Lastly, if computational resources are a concern, another option can be to use our post hoc operators which do not require any further training of the pre-trained model.

- - -

[A] Instant Neural Graphics Primitives with a Multiresolution Hash Encoding. Müller et al. SIGGRAPH 2022.

[B] Efficient Geometry-aware 3D Generative Adversarial Networks. Chan et al. CVPR 2022.

---

> ### Author Response · Authors · 2023-11-22
> **Regarding title change in the paper**
>
> Based on the Reviewer gdRC's feedback, we have made a change in the title to better reflect the problem addressed in our paper. We have incorporated this change in the revised version of the paper and will also add it to the camera-ready version.

---

### Comment · Reviewer_LQfg · 2023-11-23

As one of the reviewers advocating for this paper, I'd like to respectfully share my perspective on why I believe this paper should be considered for acceptance.

The hybrid neural field, particularly in the instantNGP style, holds a significant position within the realm of neural fields. Making enhancements within this specific class of neural fields is already very impactful and should be acknowledged. I personally think that a paper need not address all categories of neural fields to merit acceptance. I appreciate the authors' adjustment of the title to emphasize their focus on hybrid neural fields, which has led to a clearer delineation of the paper's scope.

I acknowledge the valuable contributions of the additional baselines highlighted by Reviewer DXqV, which definitely have the potential to address gradient-related challenges in hybrid neural fields. However, to the best of my knowledge, with the exception of the built-in FD, these baselines are not explicitly tailored for hybrid neural fields. Therefore, it may be somewhat unreasonable to expect the authors to provide direct comparisons with them. That said, I concur that comparing with the FD baseline is a relevant choice, given its established usage in the hybrid neural field literature, as exemplified in works such as "Neuralangelo."

The concern raised by Reviewer 6pQ5 regarding oversmoothing in PDE solvers is also valid. However, I think it's important to recognize that similar smoothing issues can manifest in classic numerical methods. For instance, in finite element methods (FEM), the choice of basis functions (whether first order or second order) can result in gradients with varying degrees of smoothness. Consequently, the smoothing issue alone should not be considered a decisive obstacle to numerically solving PDEs. On the contrary, addressing the issue of excessive noise in the original hybrid neural field is definitely a significant concern since it will definitely prevent PDE solvers from running.

In conclusion, I believe the reviewers have brought up very useful and insightful concerns. The authors should certainly consider incorporating these points into their paper, including both in discussions and conducting additional experiments where appropriate. Nevertheless, in my humble opinion, the paper has already effectively defined the scope of the problem it addresses and has provided a valid solution, including a proper comparison with relevant baselines.

---

### Meta-Review · Area_Chair_Srmk · 2023-12-05

**Metareview:**

- Claims and findings:

This paper tackles an important problem in neural fields (eg. high-frequency noise) which has  significance in computer vision (e.g., 3D reconstruction), geometry processing, and solving PDEs. The authors propose a method to accurately compute differential operators, including gradients, Hessians, and Laplacians, for hybrid grid-based neural fields. This approach effectively mitigates high-frequency noise issues associated with previous hybrid neural field techniques, particularly on high-resolution grids, by locally fitting a low-degree polynomial and computing derivative operators on the fitted polynomial function. The author's approach outperforms both auto-differentiation-based and finite difference-based methods.


- Strengths:

Reviewers have pointed out that  the angle of the paper is original and the local-polynomial fitting sounds like a simple yet effective idea to smooth out the derivatives of a neural field. In addition, reviewers have also highlighted that the paper offers a range of practical downstream applications and while it may not yield surprising results for most proposed tasks, it does exhibit improvements compared to pre-trained hybrid neural fields.

- Weaknesses:

Reviewers have noted that the claim regarding motivation from empirical observations might be somewhat overstated, particularly when applying it to all types of neural fields as opposed to only hybrid neural fields. In addition, reviewers have also pointed out that the proposed method seems to be completely based on a pre-trained network. Which means, the model is essentially being trained twice.


- Missing in submission:
N/A

**Justification For Why Not Higher Score:**

Although this submission is extremely borderline the arguments of reviewer LQfg are worth taking into account. I am leaning towards rejection since there seems to be no consensus but I wouldn't mind if the paper would get accepted as I believe it presents interesting findings.

**Justification For Why Not Lower Score:**

N/A

---

### Decision · Program_Chairs · 2024-01-16

Reject